 **eLife**

# Structural insights into human acid-sensing ion channel 1a inhibition by snake toxin mambalgin1

Demeng Sun[1,2†], Sanling Liu[1†], Siyu Li[1†], Mengge Zhang[1†], Fan Yang[1], Ming Wen[1], Pan Shi[1], Tao Wang[3], Man Pan[2], Shenghai Chang[4], Xing Zhang[4], Longhua Zhang[1*], Changlin Tian[1,3*], Lei Liu[2*]

[1]Hefei National Laboratory of Physical Sciences at Microscale, Anhui Laboratory of Advanced Photonic Science and Technology and School of Life Sciences, University of Science and Technology of China, Hefei, China; [2]Tsinghua-Peking Joint Center for Life Sciences, Ministry of Education Key Laboratory of Bioorganic Phosphorus Chemistry and Chemical Biology, Department of Chemistry, Tsinghua University, Beijing, China; [3]High Magnetic Field Laboratory, Chinese Academy of Sciences, Hefei, China; [4]School of Medicine, Zhejiang University, Hangzhou, China

**Abstract** Acid-sensing ion channels (ASICs) are proton-gated cation channels that are involved in diverse neuronal processes including pain sensing. The peptide toxin Mambalgin1 (Mamba1) from black mamba snake venom can reversibly inhibit the conductance of ASICs, causing an analgesic effect. However, the detailed mechanism by which Mamba1 inhibits ASIC1s, especially how Mamba1 binding to the extracellular domain affects the conformational changes of the transmembrane domain of ASICs remains elusive. Here, we present single-particle cryo-EM structures of human ASIC1a (hASIC1a) and the hASIC1a-Mamba1 complex at resolutions of 3.56 and 3.90 Å, respectively. The structures revealed the inhibited conformation of hASIC1a upon Mamba1 binding. The combination of the structural and physiological data indicates that Mamba1 preferentially binds hASIC1a in a closed state and reduces the proton sensitivity of the channel, representing a closed-state trapping mechanism.

**\*For correspondence:**
zlhustc@ustc.edu.cn (LZ);
cltian@ustc.edu.cn (CT);
lliu@mail.tsinghua.edu.cn (LL)

[†]These authors contributed equally to this work

**Competing interests:** The authors declare that no competing interests exist.

## Introduction

Acid-sensing ion channels (ASICs) are a group of voltage-independent proton-gated cation channels belonging to the degenerin/epithelial sodium channel (DEG/ENaC) superfamily (*Kellenberger and Schild, 2002*; *Krishtal and Pidoplichko, 1981*; *Waldmann et al., 1997*). These channels are involved in diverse physiological processes, including learning and memory (*Kreple et al., 2014*; *Wemmie et al., 2002*; *Yu et al., 2018*), neurodegeneration after ischemic stroke (*Gao et al., 2005*; *Lee and Chen, 2018*; *Qiang et al., 2018*; *Xiong et al., 2004*), and pain sensation (*Callejo et al., 2015*; *Deval et al., 2010*; *Deval et al., 2011*; *Wemmie et al., 2013*). Therefore, ASICs have emerged as potential therapeutic targets in the management of psychiatric disorders, neurodegenerative diseases and pain (*Baron and Lingueglia, 2015*; *Rash, 2017*; *Wemmie et al., 2006*; *Wemmie et al., 2013*).

Peptide toxins from venom are the most potent and subtype-selective ASIC modulators and thus have been very powerful tools for studying the gating and modulation mechanisms of ASICs (*Baron et al., 2013*; *Kalia et al., 2015*). In the past decade, structures of chicken ASIC1 (cASIC1) in different states have been reported. These include structures of the apo-form cASIC1 in the inactive (*Jasti et al., 2007*), desensitized (*Gonzales et al., 2009*) and resting (*Yoder and Gouaux, 2020*; *Yoder et al., 2018*) states and of venom-bound states: the MitTx-bound open state

(*Baconguis et al., 2014*), and the PcTx1-bound ion-selective, nonselective (*Baconguis and Gouaux, 2012*), and inactive states (*Dawson et al., 2012*). Structural studies of cASIC1-toxin complexes, combined with the structure of cASIC1 alone, have revealed a comprehensive molecular mechanism for proton-dependent gating in ASICs. In the resting state, the position of the thumb domain lies farther away from the threefold molecular axis, thereby expanding the 'acidic pocket' (*Yoder et al., 2018*). In the open and desensitized states, the 'closure' of the thumb domain into the acidic pocket expands the lower palm domain (*Baconguis et al., 2014*; *Gonzales et al., 2009*), leading to an iris-like opening of the channel gate.

Mambalgin1 (Mamba1), a 57-residue three-finger toxin isolated from the venom of the black mamba snake (*Dendroaspis polylepis polylepis),* has been proven to be a potent, rapid and reversible inhibitor of ASIC1a- or ASIC1b-containing channels in both central and peripheral neurons (*Diochot et al., 2012*). Mamba1 has an analgesic effect that is as strong as that of morphine but does not involve opioid receptors, highlighting its potential utility for the management of pain. A detailed investigation of the binding of Mamba1 to human ASICs and its resulting modulatory effect could provide insights into the mechanism of interaction between toxins and ASICs and could offer crucial clues for the development of new drugs targeting ASICs.

It should be pointed out that, to date, all ASIC structures solved and gating mechanism reported have been of chicken ASIC1, which can be pharmacologically quite different when compared to human ASIC1a despite ~89% sequence identity. Functional studies have shown that human ASIC1a (hASIC1a) and cASIC1 exhibit different responses to the channel activity modulation toxins Mamba1 (*Sun et al., 2018*). In our own experiments, synthetic Mamba1 was observed to inhibit the channel currents of both recombinant hASIC1a and cASIC1 in CHO cells (*Figure 1a*). The inhibitory effects of Mamba1 on both full-length hASIC1a and cASIC1 are concentration dependent, with $IC_{50}$ values of $197.3 \pm 37.4$ and $123.6 \pm 28.5$ nM, respectively (*Figure 1b*). Interestingly, in the presence of 500 nM Mamba1, hASIC1a and cASIC1 showed decreases of $60.4 \pm 12.9\%$ and $19.6 \pm 6.1\%$ respectively, in the measured sodium currents (*Figure 1a*). At saturation (10 µM Mamba1), hASIC1a and cASIC1 showed decreases in the measured sodium currents of $78.9 \pm 6.2\%$ and $31.9 \pm 4.7\%$ (*Figure 1b*). These data indicated that Mamba1 acts as an inhibitor targeting both hASIC1a and cASIC1, with comparable affinities but different efficacies. These observations suggested that functional and pharmacological differences exist between chicken ASIC1 and human ASIC1a, thus leading to the question of whether the structures of cASIC1 can fully recapitulate the functional states of hASIC1a. Therefore, structural studies on human ASICs are necessary to define functional states and provide comprehensive insights into the gating and toxin peptide modulation mechanisms of human ASICs.

In this study, we resolved the cryo-EM structures of hASIC1a in the apo-form and in complex with the Mamba1 toxin at 3.56 Å and 3.90 Å resolution representing the first structure of human ASICs, respectively. The structure of apo-hASIC1a was shown to be highly similar to that of cASIC1 in the resting state. Comparison of the structures of hASIC1a in the apo-form and the Mamba1-bound state revealed minor structural deviations. Electrophysiological studies revealed that Mamba1 prefers to bind hASIC1a in a closed state. Direct interactions between residues in Mamba1 and acid-sensing residues in the 'acidic pocket' of hASIC1a were observed to reduce the apparent proton sensitivity of the hASIC1a channel, leading to channel inhibition. These data indicate that the mechanism by which Mamba1 inhibits hASIC1a channel is closed-state trapping.

## Results

### Functional characterization and structure determinations

To facilitate the expression and purification of hASIC1a, a series of truncations of full-length hASIC1a was performed. Finally, truncated hASIC1a with the 60 carboxyl terminal residues removed (named hASIC1a$^{\Delta C}$) was determined to be functional by whole-cell patch-clamp electrophysiology. The hASIC1a$^{\Delta C}$ channel exhibits electrophysiological properties very similar to those of the full-length channel (*Figure 1c–d*). Mamba1 inhibits hASIC1a$^{\Delta C}$ channels with an $IC_{50}$ of $106.6 \pm 23.6$ nM, which is comparable to the reported $IC_{50}$ of full-length hASIC1a in CHO cells ($148.6 \pm 33.2$ nM) (*Figure 1b*). To better understand the structure and toxin modulation human ASICs, we set out to purify hASIC1a$^{\Delta C}$, construct the hASIC1a$^{\Delta C}$-Mamba1 complex in vitro and subject the protein and protein complex to single-particle cryo-EM studies.

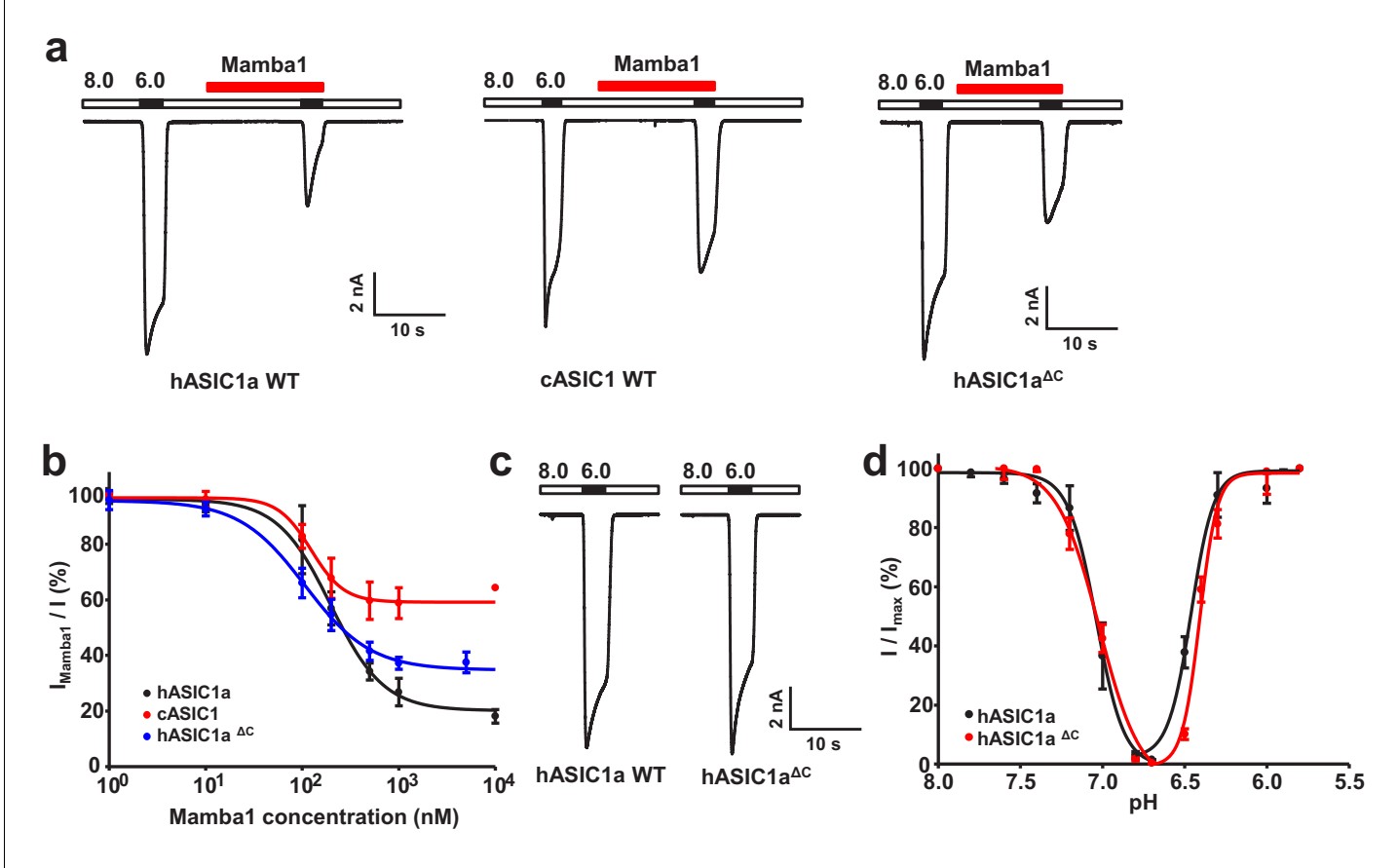

**Figure 1.** Functional analysis of hASIC1a and hASIC1a$^{\Delta C}$. (**a**) Typical current traces representing the inhibition of recombinant hASIC1a (left), cASIC1 (middle) and hASIC1a$^{\Delta C}$ (right) by Mamba1 toxin in CHO cells. (**b**) Concentration-response curve showing the inhibition of hASIC1a, hASIC1a$^{\Delta C}$ and cASIC1 expressed in CHO cells by Mamba1. $I_{mamba1}$ and I represent the currents elicited by the pH 6.0 solution in the presence and absence of Mamba1 toxin respectively. (**c**) Representative whole-cell patch-clamp recordings from wild-type hASIC1a and hASIC1a$^{\Delta C}$ activated by pH 6.0 solution. (**d**) pH-dependent activation and inactivation curves of hASIC1a (solid lines) and hASIC1a$^{\Delta C}$ (dash lines). Data were collected from CHO cells transfected with hASIC1a or hASIC1a$^{\Delta C}$ DNA. Data are presented as the mean ± SD.

The online version of this article includes the following source data and figure supplement(s) for figure 1:

**Source data 1.** Source data for *Figure 1b and d*.
**Figure supplement 1.** Purification of hASIC1a$^{\Delta C}$ for cryo-EM study.

The hASIC1a$^{\Delta C}$ protein was overexpressed and purified from *sf9* insect cells. Mamba1 was obtained through total chemical synthesis (*Fang et al., 2011*; *Fang et al., 2012*; *Pan et al., 2014*; *Schroeder et al., 2014*). The details of protein purification, complex construction, cryo-sample preparation, image acquisition, data processing, model building, and structure refinement can be found in the Materials and methods section. Briefly, recombinant hASIC1a$^{\Delta C}$ protein was purified from *sf9* cells in the presence of 0.05% (w/v) n-dodecyl-β-D-maltoside (DDM) and subjected to cryo-EM studies (*Figure 1—figure supplement 1*). Micrographs were collected on a Titan Krios electron microscope equipped with a Gatan K2 Summit detector. A 3D EM map of apo-form hASIC1a$^{\Delta C}$ was reconstructed to an overall resolution of 3.56 Å (*Figure 2—figure supplements 1–2*). Following a similar protocol, the EM map of hASIC1a$^{\Delta C}$ in complex with Mamba1 was obtained at 3.90 Å (*Figure 3—figure supplements 1–2*).

## Cryo-EM structure of apo-hASIC1a$^{\Delta C}$

The trimeric hASIC1a$^{\Delta C}$ shows a canonical chalice-like architecture (*Figure 2a–b*). Each subunit of hASIC1a$^{\Delta C}$ harbors a cysteine-rich extracellular domain (ECD). The ECD resembles a hand-like architecture with the palm, knuckle, finger and thumb domains clenching a 'ball' of β strands (*Figure 2—*

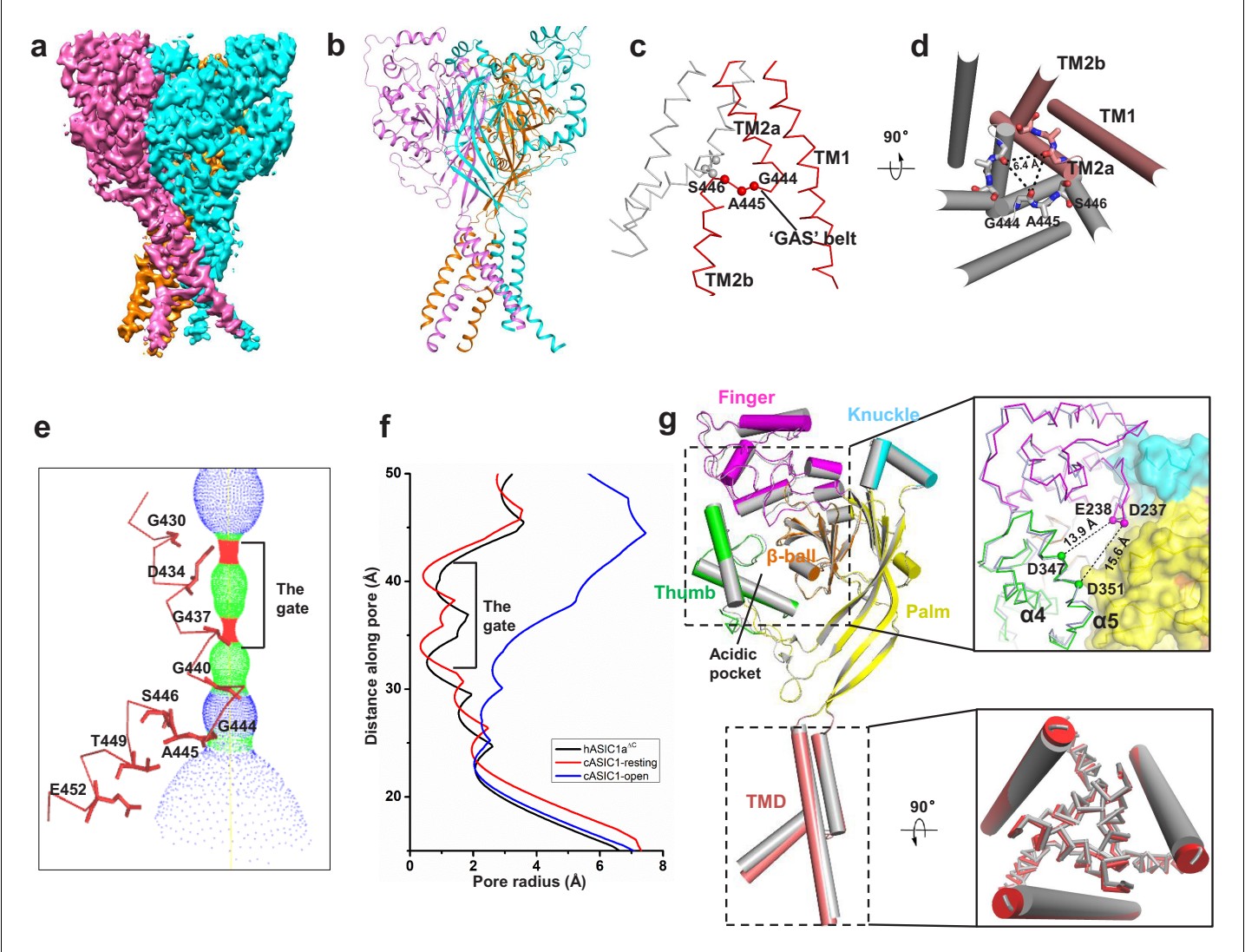

**Figure 2.** Cryo-EM structure of apo-hASIC1a$^{\Delta C}$. (a) Cryo-EM density map of apo-hASIC1a$^{\Delta C}$. The three hASIC1a$^{\Delta C}$ subunits are colored orange, cyan and pink. (b) Overall structure of trimeric hASIC1a$^{\Delta C}$, with different colors representing each subunit. (c) Ribbon representation of the hASIC1a$^{\Delta C}$ TMD. Two subunits are colored red and grey, respectively. The Cα atoms of the 'GAS belt' (G444-A445-S446) are shown as spheres. (d) View of the TMD from the intracellular side. Residues in the GAS belt are shown in stick representations. Distances between the Gly444 carbonyl oxygen atoms are indicated. (e) Close-up view of the pore domain. Map of solvent-accessible pathway is shown (red <1.4 Å<green < 2.3 Å<blue). Residues in TM2 lining the pore are shown as sticks. (f) Plot of radius as a function of longitudinal distance along the pore for hASIC1a$^{\Delta C}$ (black), cASIC1 in a resting state (red, PDB 6AVE) and cASIC1 in an open state (blue, PDB 4NTW). (g) Single subunit superposition of the apo-hASIC1a$^{\Delta C}$ and apo-cASIC1 channels in the resting state (PDB 6AVE) indicates the high similarity of the two structures. The hASIC1a$^{\Delta C}$ is represented with each domain in different colors, and cASIC1 is colored gray. The inserts show the close-up view of the acidic pocket (upper panel) and the TMD (lower panel) from the superposed hASIC1a$^{\Delta C}$ and cASIC1. For clarity, the TM1 of the TMD is shown in cartoon representation, and the TM2 is in ribbon.

The online version of this article includes the following figure supplement(s) for figure 2:

**Figure supplement 1.** Cryo-EM structure determination of hASIC1a$^{\Delta C}$.

**Figure supplement 2.** Structure model building of hASIC1a$^{\Delta C}$.

*figure supplement 2b*). In the transmembrane domain (TMD) of hASIC1a$^{\Delta C}$, two transmembrane helices, TM1 and TM2, are observed to connect to the β1 and β12 strands of the palm domain at a juncture called the 'wrist' (*Figure 2—figure supplement 2b*). The density map of TM2 provides convincing evidence that the helix is not a continuous α-helix but rather has a break in the helical structure (*Figure 2—figure supplement 2a*). The helical structure of TM2 ends at Ile443 and resumes at Ile447. Residues Gly444, Ala445, and Ser446 adopt a nonhelical conformation, dividing TM2 into

segments TM2a and TM2b. The TM2b helical element interacts with the cytoplasmic portion of TM1 of the adjacent subunit (*Figure 2c*), resulting in a swap of the TM2 helices. Overall, the three copies of TM1, TM2a and TM2b define a cavernous, threefold symmetric pore of the hASIC1a channel, in which TM2 resides on the periphery of the pore (*Figure 2d*). Electrostatic mapping of the solvent-accessible surface reveals that the ion channel pore of hASIC1a$^{\Delta C}$ harbors a modest negative potential conferred by the presence of Asp434 and Gln438, by the carbonyl oxygen atoms of Gly437, Gly440, and Gly444 (*Figure 2e*). The pore profile of hASIC1a$^{\Delta C}$ also shows a closed gate along the threefold axis as a result of primary constrictions at Asp434 and Gly437 (*Figure 2e–f*).

## The resting-state conformation of hASIC1a$^{\Delta C}$ at pH 8.0

The apo-hASIC1a$^{\Delta C}$ structure at pH 8.0 is observed to have high similarity with the resting-state structure of cASIC1 at pH 8.0 (*Yoder et al., 2018*). The two structures can be well superimposed with a root-mean-square deviation (RMSD) of 0.88 Å over 1124 aligned Cα atoms (*Figure 2g*). Especially, the structure of the acidic pockets of hASIC1a$^{\Delta C}$ and cASIC1 is almost identical, giving an RMSD of 0.41 Å. In the acidic pocket, the distances between the Cα atoms of Asp347-Glu238 and Asp351-Asp237 in hASIC1a$^{\Delta C}$ are measured to be 13.9 and 15.6 Å, respectively, which are

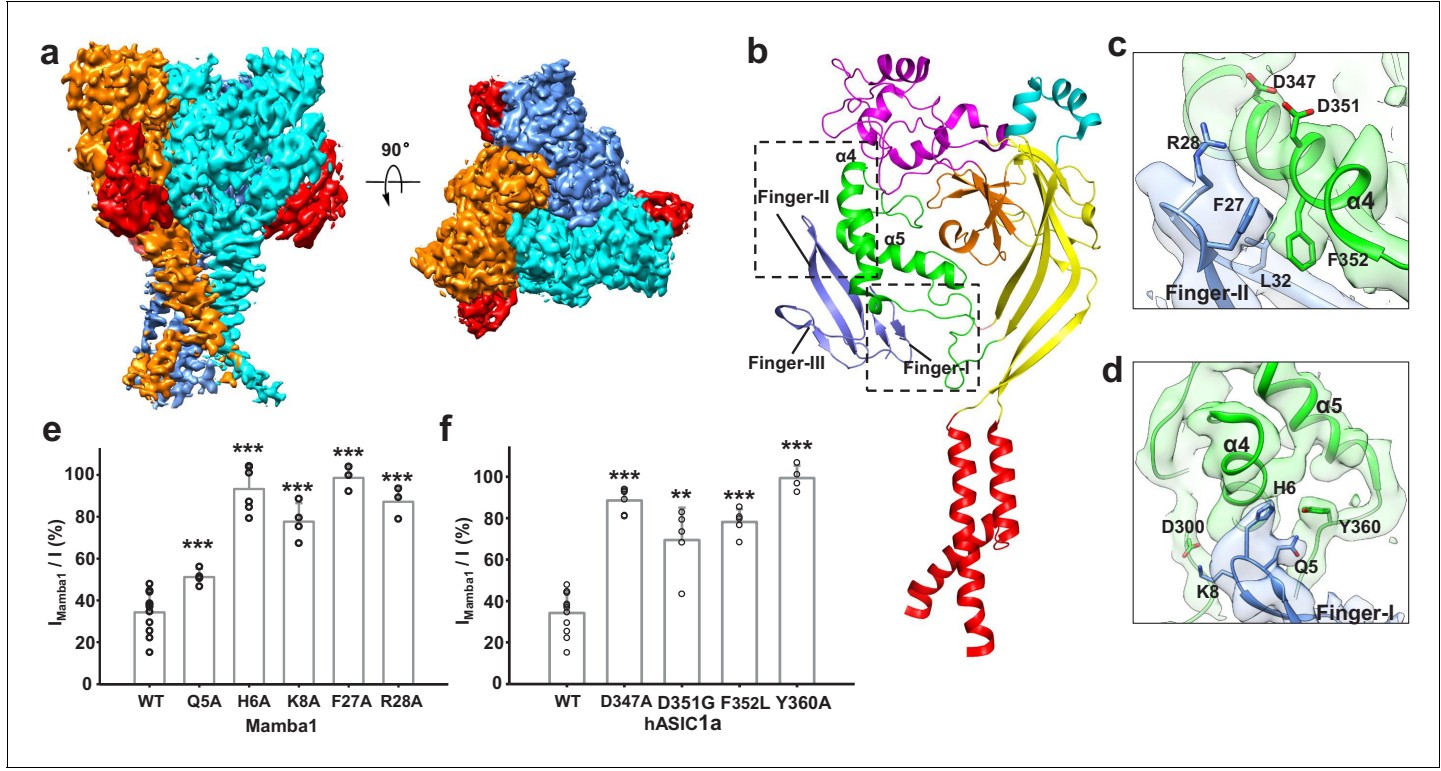

**Figure 3.** Structural basis of Mamba1 binding to hASIC1a$^{\Delta C}$. (**a**) Cryo-EM density map of the hASIC1a$^{\Delta C}$-Mamba1 complex. The three hASIC1a$^{\Delta C}$ subunits are colored orange, cyan and slate. Mamba1 is colored red. (**b**) Overall structure of the hASIC1a$^{\Delta C}$-Mamba1 complex. A single subunit of hASIC1a$^{\Delta C}$ is shown in cartoon representation, with each domain in a different color. Mamba1 is shown as a slate-colored ribbon. (**c, d**) Close-up views of the interactions of the Finger-I (**c**) and Finger-II (**d**) regions of Mamba1 with the thumb domain of hASIC1a$^{\Delta C}$. (**e, f**) Bar graph representing the inhibition of wild-type hASIC1a currents by Mamba1 mutants (**e**) and hASIC1a mutants by wild-type Mamba1 (**f**). I$_{mamba1}$ and I represent the currents elicited by the pH 6.0 solution in the presence and absence of Mamba1 toxin, respectively. Data are presented as the means ± SD. Comparison wild-type Mamba1 (**e**) or hASIC1a (**f**) unless specified, \*\*\*p<0.001; \*\*p<0.01; \*p<0.05 (t-test).

The online version of this article includes the following source data and figure supplement(s) for figure 3:

**Source data 1.** Source data for *Figure 3e and f*.
**Figure supplement 1.** Cryo-EM structure determination of hASIC1a$^{\Delta C}$-Mamba1 complex.
**Figure supplement 2.** Structure model building of hASIC1a$^{\Delta C}$-Mamba1 complex.
**Figure supplement 3.** The overlapping of the binding site of toxins on ASIC.
**Figure supplement 4.** pH-dependent activation of the hASIC1a mutants currents.
**Figure supplement 5.** One-dimensional $^{19}$F-NMR measurements of Mamba1.

comparable to the distances measured in the cASIC1 channel (13.0 Å for Asp346-Glu239 and 15.0 Å for Asp350-Asp238) (*Figure 2g*). Moreover, the structures of the TMDs of hASIC1a$^{\Delta C}$ and cASIC1 are superimposed well, with an RMSD value of 0.34 Å. The pore profiles of hASIC1a$^{\Delta C}$ and cASIC1 are almost the same (*Figure 2f*). Our structure shows that hASIC1a$^{\Delta C}$ has all the hallmarks of resting-state cASIC1, including the expanded acidic pocket, the extended GAS belt and a closed gate. Combining this information with the electrophysiology data, we conclude that the structure of hASIC1a$^{\Delta C}$ reported here represents the resting state of hASIC1a.

## Interaction between Mamba1 and hASIC1a

The overall architecture of the hASIC1a$^{\Delta C}$-Mamba1 complex shows a triskelion-like shape viewed down the threefold symmetry axis, with one Mamba1 molecule radiating from each hASIC1a$^{\Delta C}$ sub-unit (*Figure 3a*). Each of the three Mamba1 molecules binds to the ECD of a subunit of hASIC1a$^{\Delta C}$, interacting with the outside of the ECD of hASIC1a$^{\Delta C}$ (*Figure 3b*). Different with the PcTx1 and MitTx toxins, Mambal1 was observed to interact with the thumb domain of hASIC1a channel. Although the interaction between PcTx1 and cASIC1 involves the thumb, palm and β-ball domains (*Figure 3—figure supplement 3a*). The interaction between MitTx and cASIC1 involves the palm domain besides the thumb domain (*Figure 3—figure supplement 3b*).

Previously, we determined the cryo-EM structure of the cASIC1-Mamba1 complex, which revealed the binding locations of the toxin to the ECD of cASIC1 (*Sun et al., 2018*). However, the relatively low resolution (5.7 Å) could not show a reliable binding interface analysis between Mamba1 and cASIC1. Herein, the cryo-EM structure of the hASIC1a$^{\Delta C}$-Mamba1 complex at 3.9 Å resolution illuminates a more detailed binding interface between Mamba1 and hASIC1a$^{\Delta C}$. The binding of Mamba1 to hASIC1a$^{\Delta C}$ induces a conformational change in Finger-II of Mamba1. The tip region of Finger-II flips to the thumb domain of hASIC1a$^{\Delta C}$ to facilitate the interaction between the toxin and the channel (*Figure 3—figure supplement 2c*). In the tip region of Finger-II, Arg28 is oriented toward the α5 helix of the thumb domain, thus becoming closer to Asp347 and Asp351. The measured distances between the Cα atoms of Arg28-Asp347 and Arg28-Asp351 are 10.5 and 10.2 Å, respectively. Although the side chains of Arg28, Asp347 and Asp351 cannot be clearly assigned, it seems there could be salt bridges between Arg28 and Asp347 or Asp351 (*Figure 3c*). Phe352, a residue that is conserved in ASIC1 orthologues, is nestled within a hydrophobic cluster composed of Met25, Phe27, Leu32 and Leu33 in Finger-II of Mamba1, mediating a hydrophobic interaction between Mamba1 and hASIC1a$^{\Delta C}$ (*Figure 3c*). Moreover, Mamba1 could form multiple polar contacts with hASIC1a$^{\Delta C}$ through its Finger-I region. Mamba1-Gln5 and His6 could form hydrogen bonds with the side chain of Tyr360 of hASIC1a$^{\Delta C}$. Mamba1-Lys8 could form salt bridges with Asp300 (*Figure 3d*). Notably, Finger-III of Mamba1 is located farther from the thumb domain of hASIC1a$^{\Delta C}$, pointing in the opposite direction from the threefold axis of the channel core. Finger-III thus has no contact with hASIC1a$^{\Delta C}$ (*Figure 3b*). The upper scaffold region of Mamba1 likewise does not contact hASIC1a$^{\Delta C}$.

To verify the toxin-channel interactions, individual mutations were introduced into Mamba1 and the possible counterpart interaction regions on hASIC1a. The inhibition of hASIC1a activity by Mamba1 is greatly reduced when residue Mamba1-Gln5, His6, Lys8, Phe27 or Arg28 is mutated to Ala (*Figure 3e*). Mamba1 is significantly less effective in inhibiting the hASIC1a mutants Asp347Ala, Asp351Gly, Phe352Leu and Tyr360Ala (*Figure 3f* and *Figure 3—figure supplement 4*). These data are consistent with previous electrophysiological studies on chicken or rat ASIC1 channels (*Mourier et al., 2016*; *Salinas et al., 2014*; *Sun et al., 2018*).

Furthermore, the interaction of hASIC1a$^{\Delta C}$ and Mamba1 was validated using $^{19}$F-labeled Mamba1 and $^{19}$F-NMR spectroscopy in the solution state at ambient temperature. $^{19}$F labeling was introduced into Mamba1 by replacing the residue sites Phe18, Phe27, Leu30 or Leu32 in Mamba1 with $^{19}$F-labeled L-4-trifluoromethyl-phenylalanine ($^{19}$F-tfmF) through chemical synthesis. One-dimensional $^{19}$F-NMR spectra indicate a significant change in chemical shift for residues Mamba1-Phe27 (within Finger-II) upon hASIC1a$^{\Delta C}$ protein titration, and slight changes for Leu30 and Leu32, whereas no change is observed for residue Phe18 (as a negative control) (*Figure 3—figure supplement 5*). These data represent that Phe27 in Mamba1 has a significant conformational change when binds to hASIC1a channel, while the conformation of Leu30, Leu32 and Phe18 are not altered. These data indicate that Phe27 plays a key role in Mamba1 binding to hASIC1a channel.

Collectively, both mutation-based patch-clamp electrophysiology analysis and $^{19}$F-NMR measurements support the interaction between Mamba1 and hASIC1a mediated by the Finger-I/II regions of Mamba1 and the thumb domain of hASIC1a.

## The inhibited conformation of hASIC1a upon Mamba1 binding

The structure of apo-hASIC1a$^{\Delta C}$ and hASIC1a$^{\Delta C}$-Mamba1 made it possible to illustrate the detailed conformation changes in both the ECD and TMD of hASIC1a$^{\Delta C}$ upon Mamba1 binding. Structure alignment of the hASIC1a$^{\Delta C}$-Mamba1 complex and apo-hASIC1a$^{\Delta C}$ gives an RMSD of 0.44 Å$^2$, indicating the globally high similarity of the two structures (*Figure 4a* and *Figure 4—figure supplement 1a–b*). However, minor structural differences between the two structures can still be observed.

The snake toxin peptide Mamba1 contacts Asp347, Asp351, and Phe352 located in the α5 helix of the thumb domain, causing these residues to flip outward from the acidic pocket. The α5 helix thus deviates by ~5° around the central axis, whereas the α4 helix and the finger domain adopt the same conformation as in the apo-hASIC1a$^{\Delta C}$ structure (*Figure 4b*). In the hASIC1a$^{\Delta C}$-Mamba1 complex structure, the distances between the Cα atoms of Asp347-Glu238 and Asp351-Asp237 (13.3 and 12.0 Å, respectively) (*Figure 4b*) are slightly closer than those in the apo-hASIC1a$^{\Delta C}$ structure (13.9 and 15.6 Å, respectively).

In the TMD, TM1 undergoes a lateral pivot of ~6° around its carboxyl terminus upon Mamba1 binding. Meanwhile, TM2a have a shift of approximately 2.5 Å away from the channel pore, and TM2b shifts approximately 4 Å at its amino terminus and 8 Å at its carboxyl terminus away from the pore (*Figure 4c*). Comparison of the ion channel pore architectures of apo-hASIC1a$^{\Delta C}$ and the hASIC1a$^{\Delta C}$-Mamba1 complex shows that the extracellular vestibule undergoes a slight expansion upon Mamba1 binding (*Figure 4d* and *Figure 4—figure supplement 1c*). However, the overall pore profile of the hASIC1a$^{\Delta C}$-Mamba1 complex is similar to that of apo-hASIC1a$^{\Delta C}$. The transmembrane pore of the hASIC1a$^{\Delta C}$-Mamba1 complex has a diameter less than 2.0 Å in the gate, indicating a closed channel. (*Figure 4e*). The slight shifts of the thumb domain and transmembrane helices of hASIC1a in complex with Mamba1 cause it to adopt a less compact conformation than that of apo-form hASIC1a in the resting state, but the expanded conformation of the acid pocket and the closed pore are not altered.

## Mamba1 reduces the proton sensitivity of hASIC1a

Interestingly, we found that the inhibition of hASIC1a by Mamba1 depended on the pH of the toxin perfusion. When 500 nM Mamba1 was applied at conditioning pH 8.0, before the pH dropped to 6.0 (*Figure 5a*), the hASIC1a current showed decreases of 50.2 ± 8.8% at the peak (*Figure 5b*). In contrast, the coapplication of 500 nM Mamba1 as the pH dropped to 6.0 from pH 8.0 did not produce as much suppression as preapplication did (peak current showed decreases of 10.6 ± 7.2%) (*Figure 5a and b*). We also measured the current amplitudes at the end of such applications. As shown in *Figure 5b*, the currents showed decreases of 49.9 ± 19.4% when Mamba1 application was administered in the absence of the stimulating pH 6.0 application, while showed decreases of 9.3 ± 8.6% in the presence of the stimulating pH 6.0 application. These data suggest that Mamba1 binding favors a resting closed state of hASIC1a at neutral pH rather than an open or desensitized state at acidic pH.

Moreover, similar to a previous report (*Diochot et al., 2012*), in the absence of Mamba1, hASIC1a showed half-maximal activation at a pH (pH$_{50}$) of 6.46 ± 0.02. In contrast, in the presence of Mamba1 (500 nM), the pH$_{50}$ of activation was 5.89 ± 0.07 (*Figure 5c–d*). Mamba1 drastically shifts the activation curve of hASIC1a by 0.57 pH units to a more acidic pH, demonstrating that Mamba1 decreases the apparent H$^+$ affinity of hASIC1a, thus inhibiting channel activation at pH 6.0. These observations indicate that Mamba1 acts as an inhibitor targeting ASIC by modifying the proton sensitivity of the channel, consistent with previous conclusions (*Diochot et al., 2016*; *Salinas et al., 2014*).

In fact, multiple acidic residues located around the acidic pocket have been found to contribute to modulation of the proton sensitivity of ASICs, including Asp237, Glu238, Asp347 and Asp351 (*Jasti et al., 2007*). It has been reported previously that the neutralization of Asp346 and Asp350 in cASIC1 (corresponding to Asp347 and Asp351 in hASIC1a) has profound effects on either pH$_{50}$ or the apparent Hill coefficient, or both (*Jasti et al., 2007*). The Asp346Asn mutation shifts the pH

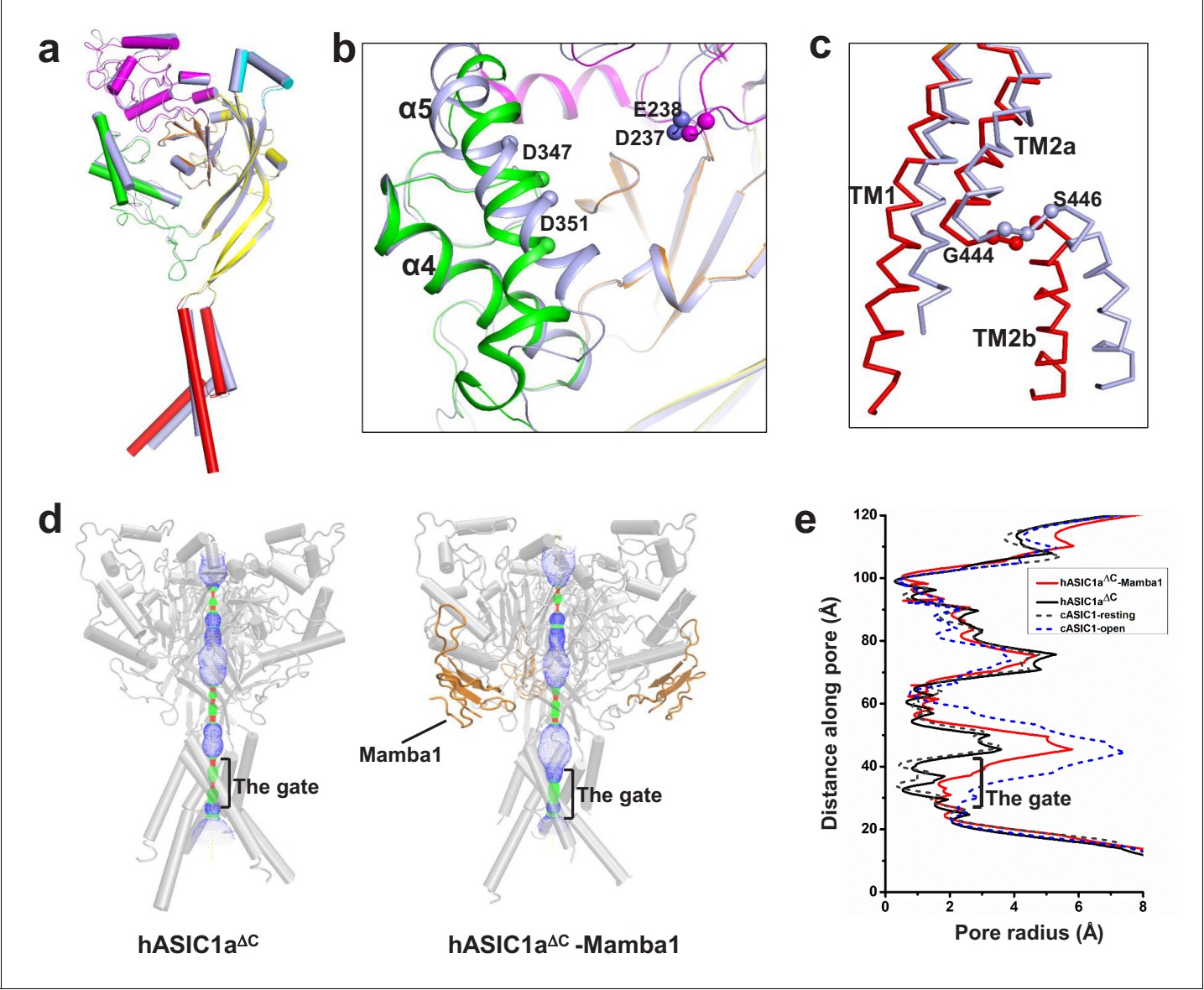

**Figure 4.** The structure of hASIC1a$^{\Delta C}$ inhibited by Mamba1. (a) Single-subunit superposition of apo form and Mamba1-bound hASIC1a$^{\Delta C}$ shows global conformational changes. The domains of apo-hASIC1a$^{\Delta C}$ are shown in different colors, and Mamba1-bound hASIC1a$^{\Delta C}$ is colored grey. (b, c) Conformational changes in the acidic pocket and TMD of hASIC1a$^{\Delta C}$ upon Mamba1 binding. (b) View of the acidic pocket from superposed apo- and Mamba1-bound hASIC1a$^{\Delta C}$. hASIC1a$^{\Delta C}$ is shown in cartoon representation and colored as in (a). Cα atoms of Glu238 and Asp347, Asp239 and Asp351 are shown as spheres. (c) View of the TMD from superposed apo- and Mamba1-bound hASIC1a$^{\Delta C}$. The TMDs of hASIC1a$^{\Delta C}$ and the hASIC1a$^{\Delta C}$-Mamba1 complex are shown as ribbons in red and grey, respectively. (d) Pore profiles of hASIC1a$^{\Delta C}$ (left) and the hASIC1a$^{\Delta C}$-Mamba1 complex (right) calculated with HOLE software (red <1.4 Å<green < 2.3 Å<blue). (e) Plot of pore radius for the apo-form hASIC1a$^{\Delta C}$ (black, solid line), hASIC1a$^{\Delta C}$-Mamba1 complex (red, solid line), cASIC1 in resting state (grey, dash line, PDB 6AVE), and cASIC1 in open state (blue, dash line, PDB 4NTW) along the threefold molecular axis.

The online version of this article includes the following figure supplement(s) for figure 4:

**Figure supplement 1.** Structure comparison of apo-hASIC1a$^{\Delta C}$ and hASIC1a$^{\Delta C}$-Mamba1 complex.

dependence of cASIC1 activation to more acidic values (6.35 ± 0.04 to 5.58 ± 0.02) and reduces the Hill coefficient from approximately 9 to 5, and the Asp350Asn mutation diminishes the Hill coefficient to approximately three and has little effect on pH$_{50}$ (*Jasti et al., 2007*). Our data show that hASIC1a-Asp347Ala and hASIC1a-Asp351Gly both shift the pH-dependent activation curves to more acidic pHs (*Figure 5c–d*). The Asp347Ala and Asp351Gly mutations shift the activation curve of

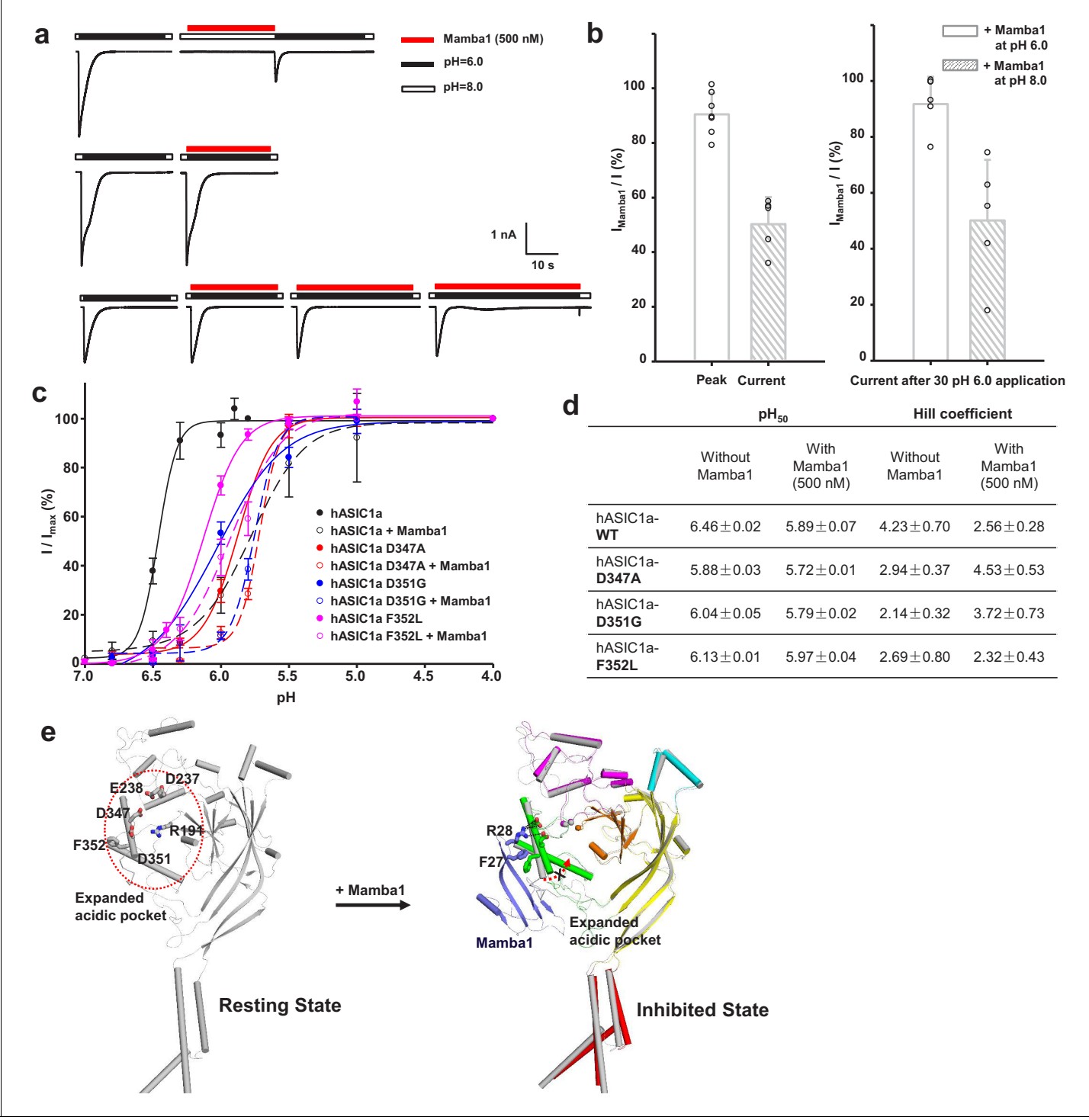

**Figure 5.** The 'closed-state trapping' model for Mamba1 inhibition of hASIC1a activity. (**a**) Typical traces of hASIC1a currents recorded in CHO cells with administrations of Mamba1 (500 nM). The Mamba1 toxin applications were administered either in the absence (upper panel) or presence (middle and lower panels) of the stimulating pH 6.0 application. (**b**) Bar graph representing the inhibition of the peak currents of hASIC1a statistics with administrations of Mamba1 at pH 6.0 and pH 8.0 (left panel). The inhibition of current amplitudes at the end of such applications are also measured and statistically compared (right panel). (**c**) pH-dependent activation curves of wild-type hASIC1a (WT) and hASIC1aΔmutants obtained in the absence or presence of Mamba1. (**d**) The measured $pH_{50}$ and Hill coefficient values of wild-type hASIC1a (hASIC1a-WT) and hASIC1a mutants in the conditions with or without Mamba1. Data are presented as the mean ± SEM. (**e**) The closed-state trapping inhibition mechanism of hASIC1a by Mamba1. Mamba1 binding leads to deformation of the proton-sensitive site, then stabilizes the expanded conformation of the acidic pocket and traps the channel in a

*Figure 5 continued on next page*

Figure 5 continued

closed state. The structure of a single subunit of hASIC1a$^{\Delta C}$ is shown as a cartoon in gray, representing the resting state of the channel (left). The structure of hASIC1a$^{\Delta C}$-Mamba1 is shown as a cartoon, with each domain colored differently. Mamba1 is colored blue (right).

The online version of this article includes the following source data for figure 5:

**Source data 1.** Source date for *Figure 5b, c and d*.

hASIC1a by 0.59 and 0.42 pH units, respectively, with $pH_{50}$ values of $5.88 \pm 0.03$ and $6.04 \pm 0.05$. The mutations reduce the Hill coefficient from approximately 4 to 2. Meanwhile, Mamba1 is able to shift the activation curves of the hASIC1a-Asp347Ala and hASIC1a-Asp351Gly mutants further, with $pH_{50}$ values of $5.72 \pm 0.01$ and $5.79 \pm 0.02$, respectively (*Figure 5c–d*). We suggest that probable interactions between Asp347 and/or Asp351 of hASIC1a and Mamba1 shield the protonation of the two residues, thus reducing the proton sensitivity of hASIC1a. On the other hand, the interaction between Asp351 and Mamba1 could hinder the interaction between Asp351 and Arg190 (located at the β-ball), which is critical for the collapse of the acidic pocket in active or desensitized channels. Therefore, we suspect that Mamba1 inhibits the hASIC1a channel by hindering proton-induced transitions from the resting closed states to the active and/or desensitized states in a model of closed-state trapping (*Figure 5e*).

Notably, Asp347 and Asp351 are not the only proton-sensing residues in hASIC1a (*Vullo et al., 2017*). Glu79, Glu219, Glu409 and Glu418, which are located in the palm domain, which have no direct contact with Mamba1, were also found to participate either in proton binding or in subsequent conformational changes (*Jasti et al., 2007*; *Krauson et al., 2013*; *Ramaswamy et al., 2013*). This observation may explain why Mamba1 can inhibit only approximately 80% of the hASIC1a current and loses its inhibitory effect on hASIC1a at a lower pH, such as pH 4.0.

## The intradomain interaction affects the Mamba1 inhibition of ASIC

As shown in *Figure 1*, although Mamba1 has similar affinity to hASIC1a and cASIC1 ($IC_{50} = 197.3 \pm 37.4$ and $123.6 \pm 28.5$ nM, respectively), the toxin shows a stronger inhibitory effect on hASIC1a than on cASIC1 channels. In fact, the amino acid sequences of hASIC1a and cASIC1 are highly similar (89% identity), and the sequences of the thumb domain of the two orthologues are identical (*Figure 6—figure supplement 1*). This similarity may explain the similar affinity of Mamba1 in targeting hASIC1a and cASIC1. Comparison of the apo-form structure of hASIC1a with the resting-state structure of the cASIC1 channel, together with the structure of the hASIC1a-Mamba1 complex, showed comparable expansion of the acid pocket between chicken and human ASIC1a channels (*Figures 2g* and *4b*). The key residues (such as Asp346 and Phe351) on the α5 helix in the thumb domain of chicken ASIC1 channels that participate in Mamba1 binding are also critical in human ASIC1a channels. These observations suggest that the different inhibitory efficacies of Mamba1 on cASIC1 and hASIC1a channels could not be due to the toxin-channel interaction.

Through sequence alignment analysis and structure comparison, several key residues that do not contribute to Mamba1 binding were found to contribute to the different inhibitory effects of Mamba1 on hASIC1a and cASIC1. These residues include Gln102, Arg155 (both located in the finger domain of hASIC1a) and Asp167 (located in the palm domain), which are mapped to Arg103, Leu156 and Glu168 of cASIC1, respectively (*Figure 6a and b*). In cASIC1, Leu156 is observed to interact with the hydrophobic pocket composed of Phe189, Val327 and Tyr334 in the thumb domain (*Figure 6c*, *Figure 6—figure supplement 2*). The hydrophobic contact could participate in the interaction between the finger and thumb domains. Meanwhile, Arg103 and Glu168 could form electrostatic interaction pairs with each other (the distance between the Cα atom of the two residues was measured to be 13.3 Å), mediating the contact between the finger and palm domains (*Figure 6d*, *Figure 6—figure supplement 2*). It was reported that the movement of the finger domain plays a critical role in the activation of ASICs (*Bonifacio et al., 2014*; *Gwiazda et al., 2015*; *Krauson and Carattino, 2016*; *Ramaswamy et al., 2013*; *Vullo et al., 2017*). The replacement of Leu156 by Arg155 and of Arg103-Glu168 by Gln102-Asp167 in hASIC1a result in disruption of intradomain interactions, thus leading to enhanced inhibition of the channel by Mamba1.

To elucidate the contributions of these residues to the inhibition of ASIC1 activity by Mamba1, mutations were introduced at these sites in hASIC1a channels, and whole-cell patch-clamp

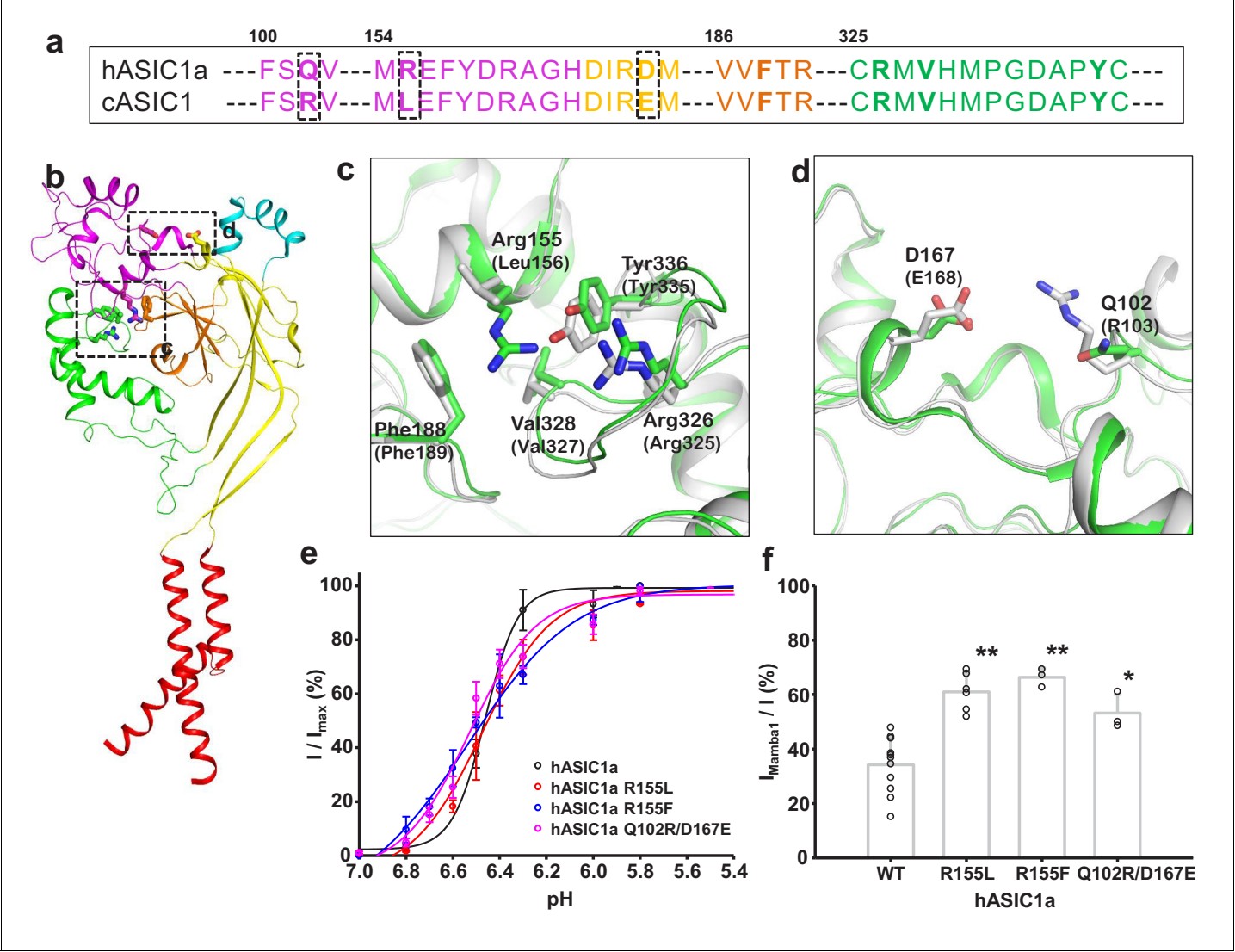

**Figure 6.** Structural basis for the differing activities of hASIC1a versus cASIC1. (**a**) Sequence alignment of hASIC1a and cASIC1 indicates key residues that may contribute to the inhibitory effect of Mamba1 on ASIC. (**b**) An individual subunit of resting-state hASIC1a$^{\Delta C}$. (**c**, **d**) Local alignments of hASIC1a$^{\Delta C}$ (green) with resting-state cASIC1 (grey, PDB 5WKV) demonstrate the possible interactions of residues in two handles. (**e**) pH-dependence curves of hASIC1a and its mutants. (**f**) Bar graph representing the inhibitory effect of wild-type Mamba1 (500 nM) on hASIC1a mutants. Data are presented as means ± SD. Comparison with wild-type hASIC1a, **p<0.01; *p<0.05 (t-test).

The online version of this article includes the following source data and figure supplement(s) for figure 6:

**Source data 1.** Source data for *Figure 6e and f*.
**Figure supplement 1.** Sequence alignment of chicken ASIC1 and human ASIC1a channels.
**Figure supplement 2.** The density maps of the key residues that possibly mediate domain interactions in hASIC1a.

electrophysiology analysis were performed. As expected, replacing Arg155 of hASIC1a with Leu or Phe decreased the inhibition of the channel activity by Mamba1 but did not affect the proton sensitivity of the channel. The same was true for the Arg and Glu substitutions at hASIC1a-Gln102 and Asp167, respectively (*Figure 6e–f*). These observations support the hypothesis that intrasubunit interactions play critical roles in the activity modulation of ASICs from different species by toxins.

## Discussion

To date, many electrophysiological studies of ASIC1 channels have been based on mice, rats and chickens. Crystal or cryo-EM structures of chicken ASIC1 in different states were reported as

illustrating the gating mechanism of the channel (*Baconguis et al., 2014*; *Baconguis and Gouaux, 2012*; *Gonzales et al., 2009*; *Jasti et al., 2007*; *Sun et al., 2018*; *Yoder and Gouaux, 2020*; *Yoder et al., 2018*). Here, we report the first structure of the human ASIC1a channel determined using single-particle cryo-EM with a resolution of 3.56 Å.

The structure of hASIC1a in the apo-form at pH 8.0 reflects the resting state of the channel. hASIC1a in the resting state has an identical structure to that of the cASIC1 channel, especially the expanded acidic pocket and the extended 'GAS' motif. In fact, the human ASIC1a channel and chicken ASIC1 channel have high-sequence homology (89% identity). These indicate that hASIC1a and cASIC1 channels may share a similar gating mechanism. That is, the resting-state hASIC1a channel has an expanded acidic pocket, with the thumb domain far away from the central β-ball and the finger and palm domains. Activation and desensitization of the channel both involve the collapse of the acidic pocket, which allows the thumb and finger domains to approach and interact with each other (*Gonzales et al., 2009*; *Baconguis et al., 2014*; *Yoder et al., 2018*). However, the structures of human ASIC in the active or desensitized state remain unknown. Resolving the structures of the active or desensitized hASIC1a will help to address this supposition.

In recent decades, venom toxin peptides have been observed to bind ASICs with high affinity and specificity, providing an excellent resource for the definition of different functional states of the channel (*Bohlen et al., 2011*; *Diochot et al., 2004*; *Diochot et al., 2012*; *Escoubas et al., 2000*; *Reimers et al., 2017*). We also report here the structure of the human ASIC1a channel in complex with the snake peptide toxin Mamba1 with resolution 3.90 Å, reflecting the toxin-inhibited state of the channel. A first important conclusion is that, the structure of the hASIC1a-Mamba1 complex confirms that the Mamba1 binding site is located in the thumb domain of the hASIC1a channel. Compared with the previously reported structures of the cASIC1 channel in complex with toxins PcTx1, MitTx or Mamba1, these toxins interact with the ASIC channel through overlapping, but not identical surfaces. The thumb domain is a hot spot that mediates toxin-channel interactions. However, the Mamba1-binding site is totally located in the thumb domain of the hASIC1a channel, and does not involve other sub-domains. In contrast, the toxin-binding surfaces for MitTx and PcTx1 involve the β-ball and the palm domain in addition to the thumb domain. These observations could explain the fact that MitTx and PcTx1 have much higher affinity than Mamba1 targeting ASIC channels.

Moreover, structure comparison of hASIC1a in the apo-form and in complex with Mamba1 reveals that Mamba1 binding does not alter the closed structure of hASIC1a in a resting state. Therefore, we conclude that Mamba1 inhibits hASIC1a through a closed-state trapping mechanism, precluding the previously proposed allosteric-based channel modulation mode. Furthermore, the binding of Mamba1 to hASIC1a is state-dependent. The toxin preferentially binds to the closed state but not the active or desensitized state of ASICs. Mamba1 inhibits ASICs by shifting the pH-dependence of activation to a more acidic pH, decreasing their apparent affinity for protons. In fact, the state-dependent trapping mechanism has been found in the modulation of voltage-gated ion channel activity by peptide toxins. It is proposed that voltage sensor trapping is the fundamental mechanism of action of polypeptide toxins that alter the voltage-dependent gating of sodium, calcium, and potassium channels (*Cestèle et al., 1998*). For the ASIC channel, the two Asp residues in the α5 helix of the thumb domain contribute to proton sensing during channel activation (*Jasti et al., 2007*). Therefore, the thumb domain could act as the 'proton sensor' of the channel. Accordingly, we conclude that Mamba1 toxin inhibits ASIC channel activity through a closed-state dependent 'proton sensor trapping' mechanism, sharing a common mechanism of polypeptide toxins modulating ion channel activity.

Mamba1 showed a stronger effect on hASIC1a than on cASIC1. Previous reported key residues (such as Asp346 and Phe351) on the α5 helix in the thumb domain of the chicken ASIC1 channel that participated in Mamba1 binding were also critical in the human ASIC1a channel. The structural analysis and mutation experiments suggested that the different inhibitory effects of Mamba1 targeting cASIC1 and hASIC1a channels might be due to the mechanism of channel activation rather than the difference in toxin-channel interactions. Several key amino acid differences between hASIC1a and cASIC1, including Arg155 and the Glu102-Asp165 pair in hASIC1a (corresponding to Leu156 and Arg103-Glu168 in cASIC1, respectively), are found to contribute to the different responses of hASIC1a and cASIC1 to Mamba1. Due to the relatively low resolution, our structures reported here could not provide detailed structural evidence sufficient to support this finding. It is important to

address how the coupling among the extracellular subdomains of ASIC, including the thumb, finger, palm and β-ball, affects channel activity and toxin action on the channel.

ASIC channels are of fundamental importance and are also considered potential drug targets in therapeutic interventions against pain and ischemic stroke. There is no doubt that analysis of interactions between venom toxin and hASIC1a or peptide toxin-based drug development should follow structure and function studies on human source target proteins, as data on homologous target proteins from other species might not precisely reflect the modulatory effects of the peptide or of other ligands on the target proteins, which will strongly influence the process of drug development for human target proteins. Our studies on the human ASIC1a channel could obviously be very valuable for drug development targeting ASIC.

# Materials and methods

## Key resources table

| Reagent type (species) or resource | Designation | Source or reference | Identifiers | Additional information |
|---|---|---|---|---|
| Gene (*Homo sapiens*) | hASIC1a | GeneBank | NCBI Reference Sequence: NP_001086.2 | All hASIC1a mutants transfected in the paper were obtained starting from this wild-type cDNA |
| Strain, strain background (*Escherichia coli*) | Top10 | Thermo Fisher Scientific | Cat# 18258012 | |
| Strain, strain background (*Escherichia coli*) | DH10Bac | Thermo Fisher Scientific | Cat# 10361012 | |
| Cell line (*Spodoptera frugiperda*) | Sf9 | Thermo Fisher Scientific | Cat# 11496015; RRID:CVCL_0549 | |
| Cell line (*Homo sapiens*) | HEK-293T | ATCC | Cat#: CRL-3216; RRID:CVCL_0063 | |
| Cell line (*Cricetulus griseus*) | CHO-K1 | ATCC | Cat# 03480/ p693_CHO-K1; RRID:CVCL_0214 | |
| Recombinant DNA reagent | pFastBac1 | Invitrogen | | |
| Recombinant DNA reagent | pcDNA3.1 | Invitrogen | | |
| Chemical compound, drug | n-Dodecyl-β-D-Maltoside (DDM) | Anatrace | Cat#: D310 | |
| Chemical compound, drug | Cholesterol Hemisuccinate tris Salt (CHS) | Sigma-Aldrich | Cat#: C6013 | |
| Peptide, recombinant protein | Mamba1 | This paper | UniProtKB: P0DKR6 | Mamab1 and mutants were obtained by chemical synthesis |
| Software, algorithm | Gctf | *Zhang, 2016* | https://www2.mrc-lmb.cam.ac.uk/research/locally-developed-software/zhang-software/#gctf | |
| Software, algorithm | RELION 3.1 | *Zivanov et al., 2018* | http://www2.mrclmb.cam.ac.uk/relion; RRID:SCR_016274 | |

*Continued on next page*

*Continued*

| Reagent type (species) or resource | Designation | Source or reference | Identifiers | Additional information |
|---|---|---|---|---|
| Software, algorithm | SerialEM | *Mastronarde, 2005* | RRID:SCR_017293 | |
| Software, algorithm | PHENIX | *Liebschner et al., 2019* | https://www.phenixonline.org; RRID:SCR_014224 | |
| Software, algorithm | Coot | *Emsley et al., 2010* | https://www2.mrc-lmb.cam.ac.uk/personal/pemsley/coot; RRID:SCR_014222 | |
| Software, algorithm | UCSF Chimera | *Pettersen et al., 2004* | https://www.cgl.ucsf.edu/chimera; RRID:SCR_004097 | |
| Software, algorithm | PyMol | Schrödinger | https://pymol.org/2; RRID:SCR_000305 | |
| Software, algorithm | GraphPad Prism 7 | GraphPad Software | https://www.graphpad.com/scientific-software/prism | |
| Software, algorithm | HOLE | *Smart et al., 1996* | http://www.holeprogram.org | |
| Others | QUANTIFOIL R1.2/1.3 holey carbon grids | Quantifoil | | |
| Others | Cellfectin | Invitrogen | Cat# 10362100 | |
| Others | Superose 200 Increase 10/300 GL | GE Healthcare | Cat# 28990944 | |

## Cell lines

All cell lines used were obtained from commercial sources (see the Key Resources Table). Sf9 cells were cultured at 27°C in serum-free SIM SF medium (Sino Biological Inc). HEK293T cells were cultured as adherent cells in DMEM (with L-glutamine, glucose and sodium pyruvate), supplemented with 10% FBS and 1% Gibco antibiotic-antimycotic; at 37°C in 5% $CO_2$. CHO-K1 cells were cultured in DMEM/F12 medium (Gibco) supplemented with 10% fetal bovine serum (FBS), 100 U/mL penicillin, and 100 U/mL streptomycin at 37°C in a 5% $CO_2$ incubator. No additional authentication was performed by the authors of this study. Cell line was negative for mycoplasma. No commonly misidentified lines were used in this study. All cell lines were kept at low passages in order to maintain their health and identity.

## Protein expression, purification and complex construction

The mambalgin1 (Mamba1) toxin was obtained using a hydrazide-based chemical synthesis method as previously reported (*Pan et al., 2014*). The polypeptide chain of Mamba1 (57 amino acids) was divided into three segments at two ligation sites (Cys19 and Cys41). All the segments (Mamba1[1-18]-NHNH$_2$, Mamba1[19–40]–NHNH2 and Mamba1[41–57]) were synthesized using a standard solid-phase peptide synthesis method. The three segments were then ligated through the standard hydrazide-based native chemical ligation (NCL) to synthesize the full-length Mamba1. This synthesis of Mamba1 is convenient and produced high yields following the final HPLC purification (35% isolated yield, multi-milligram scale and good homogeneity). For the synthesis of Mamba1 mutants, the Alanine or $^{19}$F-labeled L-4-trifluoromethyl-phenylalanine (19F-tfmF) were incorporated directly during the peptide segments synthesis.

The optimized coding DNAs for human hASIC1a (Uniprot: P78348) was synthesized by GeneScript. The truncated hASIC1a (with the carboxyl terminal 60 residues removed, named as hASIC1a$^{\Delta C}$) was cloned into the pFastBac1 vector (Invitrogen) with 8-His tag at the amino terminus. Baculovirus-infected *sf9* cells (Thermo Fisher) were used for overexpression and were grown at 27°C in serum-free SIM SF medium (Sino Biological Inc). Cells were harvested 2 days after infection by

centrifugation at 1000 g and resuspended in lysis buffer containing 20 mM Tris (pH 8.0), 200 mM NaCl for each batch of protein purification. The suspension was supplemented with 1% (w/v) n-dodecyl-β-D-maltopyranoside (DDM, Anatrace), 0.2% (w/v) cholesteryl hemisuccinate Tris salt (CHS, Anatrace) and protease inhibitor cocktail (Sigma). After incubation at 4°C for 2 hr, the solution was ultracentrifuged at 200,000 g for 45 min, and the supernatant was applied to Ni-NTA (GE HealthCare) by gravity at 4°C. The resin was rinsed four times with the buffer containing 20 mM Tris (pH 8.0), 200 mM NaCl, 40 mM imidazole, 0.1% DDM, 0.02% CHS and the protease inhibitor cock-tail. The target proteins were eluted with buffer containing 20 mM Tris (pH 8.0), 200 mM NaCl, 250 mM imidazole, 0.1% DDM, 0.02% CHS. The eluted protein was further purified by size-exclusion chromatography in 20 mM Tris (pH 8.0), 200 mM NaCl, 0.05% DDM, 0.01% CHS using a Super-dex200 10/300 GL column (GE HealthCare). The presence of hASIC1a$^{\Delta C}$ in the peak fractions of size exclusion chromatography purification was confirmed by SDS-PAGE and mass spectrometry (MS).

To construct the hASIC1a$^{\Delta C}$-Mamba1 complex, hASIC1a$^{\Delta C}$ was purified as described above in pH 8.0 buffer and concentrated to about 5 mg/ml based on $A_{280}$ measurement, using a 100 kDa cutoff Centricon (Millipore). The chemical synthesized, lyophilized Mamba1 was dissolved in buffer contain-ing 20 mM Tris (pH 8.0), 200 mM NaCl, 0.05% DDM, 0.01% CHS at a final concentration of 10 mg/ml based on $A_{280}$ measurement, and added in a 6:1 molar ratio of toxin to channel with incubation for 1 hr at 4°C.

## Single-particle cryo-EM data acquisition

Purified hASIC1aΔC (3 μl) at a concentration of 2.7 mg/ml was added to the freshly plasma-cleaned holey carbon grid (Quantifol, R1.2/1.3, 300 mesh, Cu), blotted for 5 s at 100% humidity with a Vitro-bot Mark IV (ThermoFisher Scientific) and plunge frozen into liquid ethane cooled by liquid nitrogen. Cryo-EM sample of hASIC1a$^{\Delta C}$-Mamba1 complex was prepared similarly with the concentration of 3.1 mg/ml. Grids were transferred to a Titan Krios electron microscope (FEI) operated at 300 kV equipped with a Gatan K2 Summit direct detection camera. Images of hASIC1a$^{\Delta C}$ and hASIC1a$^{\Delta C}$-toxin complexes were collected using the automated image acquisition software SerialEM in count-ing mode with 29,000 × magnification yielding a pixel size of 1.014 Å. The total dose of 50 e$^-$/Å$^2$ was fractionated to 40 frames with 0.2 s per frame. Nominal defocus values ranged from −1.8 to −2.5 μm. The datasets of hASIC1a$^{\Delta C}$ and hASIC1a$^{\Delta C}$-Mamba1 complex included 3235 and 3364 micrographs, respectively.

## Image processing

Dose-fractionated image stacks were subjected to beam-induced motion correction and dose-weighting using UCSF MotionCor2 (*Zheng et al., 2017*). Contrast transfer function parameters were estimated with Gctf (*Zhang, 2016*). For particle picking, 2000 particles were picked manually to gen-erate references for autopicking. The atuopicked particles were extracted by four-times downscaling resulting in the pixel size of 4.056 Å and then subjected to reference-free 2D classification in Relion-2 (*Kimanius et al., 2016*). For the dataset of hASIC1a$^{\Delta C}$, 213,605 particles from well-defined 2D averages were selected for 3D classification with a pixel size of 2.028 Å. A 3D initial model de novo from the 2D average particles was generated using stochastic gradient descent (SGD) algorithm in Relion-2. The 50 Å low-pass filtered initial model was used as a template for 3D classification into four classes. A selected subset of 122,890 particles were used to obtain the final map with a pixel size of 1.014 Å and C3 symmetry imposed in the last round of 3D refinement in Relion-2. The global resolution of this map was estimated to be 3.56 Å based on the gold-standard Fourier shell correla-tion (FSC) using the 0.143 criterion. The dataset of hASIC1a$^{\Delta C}$-Mamba1 complex was similarly proc-essed in Relion, with a subset of 119,901 particles producing a final map with global resolution of 3.90 Å. Local resolution was determined using ResMap (*Kucukelbir et al., 2014*) with unfiltered half-reconstructions as input maps.

## Model building

The coordinate of chicken ASIC1 (PDB code 6AVE) (*Yoder et al., 2018*) was fitted into the 3D EM maps of hASIC1a$^{\Delta C}$ using UCSF Chimera (*Pettersen et al., 2004*). The sequence of cASIC1 were mutated with corresponding residues in human ASIC1a in Coot (*Emsley et al., 2010*). Every residue was manually examined. The chemical properties of amino acids were considered during model

building. The N-terminal residues 1–39 and C-terminal residues 466–468 were not built due to the lack of corresponding densities. Structure refinement and model validation were performed using phenix.real_space_refine module in PHENIX (*Adams et al., 2010*; *Afonine et al., 2018*). The refined model of hASIC1a$^{\Delta C}$ and the coordinate of Mamba1 (PDB code 5DU1) (*Mourier et al., 2016*) were fitted into the 3D map of hASIC1a$^{\Delta C}$-Mamba1 complex in Chimera. All the residues were manually adjusted in Coot. The final model was subjected to refinement and validation in PHENIX.

## Plasmid construction, cell culture and transient transfection of CHO cells

The coding sequence for wild-type hASIC1a was sub-cloned into the pcDNA3.1/Zeo(+) vector. All site-directed mutations were generated with overlap PCR and inserted into pcDNA3.1/Zeo(+). The mutants were sequenced to verify that no unwanted mutations had been introduced. Chinese hamster ovary (CHO) cells were cultured in DMEM/F12 medium (Gibco) supplemented with 10% fetal bovine serum (FBS), 100 U/mL penicillin, and 100 U/mL streptomycin at 37°C in a 5% $CO_2$ incubator. The CHO cells were transferred to 24-well plates for transfection. When the CHO cells reached 90% confluence, they were transfected with 0.6 μg of plasmid encoding EGFP and 0.8 μg of plasmid encoding wild-type or mutant hASIC1a using Lipofectamine 2000 (Invitrogen, USA). After incubation for 5 hr, the cells were transferred to poly-L-lysine (Sigma)-coated slides for culture for another 24–48 hr in fresh medium. They were then used for the electrophysiological analysis.

## Electrophysiological analysis of CHO cells

For the whole-cell recordings, the bath solution contained 150 mM NaCl, 4 mM KCl, 2 mM $CaCl_2$, 1 mM $MgCl_2$, and 10 mM HEPES (pH 8.0,~308 mOsm). The electrodes were pulled from thick-walled borosilicate glass capillaries with filaments (1.5 mm diameter; Sutter Instruments) on a four-stage puller (P-1000; Sutter, USA) and had resistances of 2–5 MΩ when filled with intracellular solution containing 140 mM KCl, 10 mM NaCl, 5 mM EGTA, 10 mM HEPES, (pH 8.0,~297 mOsm). All chemicals were obtained from Sigma. The experiments were performed at room temperature with an EPC-10 amplifier (HEKA Electronic) with the data acquisition software PatchMaster. Membrane potential was held at −70 mV in all experiments. Acid-induced currents were recorded by rapidly exchanging local solution from pH 8.0 to acidic pH through a Y-tube perfusion system. Toxins were applied 30 s before the pH decreased and persisted during low pH application. Channels were activated by acid perfusion at least every 2 min to allow for a complete recovery of the channels from desensitization. Recordings in which access resistance or capacitance changed by 10% during the experiment were excluded from data analysis. Mamba1 was added when the currents were stable.

## Patch-clamp electrophysiological data analysis

The data were analyzed with Clampfit and SigmaPlot. The dose–response curves used to determine the IC$_{50}$ values were fitted using the Hill equation: y = 1 + (Imax − 1)/(1 + (IC$_{50}$/x)h), where x is the toxin concentration, h is the Hill coefficient, and IC$_{50}$ is the half-maximal effect. The results are presented as the means ± standard errors (SE), and n is the number of experiments.

## $^{19}$F NMR spectra measurements

All one-dimensional $^{19}$F NMR spectra measurements were performed at 298 K on a Bruker 600MHz spectrometer equipped with a triple inverse TCI cryo-probehead, H and F-C/N-D-05-Z probe and the observation channel was tuned to $^{19}$F (564.7 MHz), with 10240 free induction decay (FID) accumulations in every 1 s recycling delay, 4096 scans per experiment. One-dimensional 19F spectra were acquired with one pulse program with 90° pulse width of 11 μs and power at 7.09 w. The $^{19}$F chemical shifts were calibrated using the internal standard TFA. The data were processed and plotted with an exponential window function (line broadening = 20 Hz) using TopSpin 4.0.5. The concentration of tfmF site-specific-labeled Mambalgin-1 was 0.1 mM containing 10% $D_2O$. Finally, 0.1 mM 19F-labeled Mambalgin-1 was added into 0.1 mM hASIC1a containing 10% $D_2O$ in 400 μl.

## Acknowledgements

We thank Dr. Linfeng Sun (University of Science and Technology of China) for technical support during data processing. We thank the Center for Integrative Imaging of Hefei National Laboratory for Physical Sciences at the Microscale of University of Science and Technology of China (Hefei), and the Center of Cryo-Electron Microscopy of Zhejiang University (Hangzhou) for providing the cryo-EM facility support. Funding: This work was funded by the National Key Research and Development Project (2016YFA0400903, 2017YFA0505201 and 2017YFA0505403), National Natural Science Foundation of China (31600601, 21778051 and 91753205), Queensland-Chinese Academy of Sciences (Q-CAS) Collaborative Science Fund for Changlin Tian and Glenn King (GJHZ201946).

## Additional information

### Funding

| Funder | Grant reference number | Author |
|---|---|---|
| National Key Research and Development | 2017YFA0505201 | Lei Liu |
| National Natural Science Foundation of China | 31600601 | Changlin Tian |
| National Natural Science Foundation of China | 91753205 | Lei Liu |
| National Key Research and Development | 2016YFA0400903 | Demeng Sun |
| National Key Research and Development | 2017YFA0505403 | Lei Liu |
| National Natural Science Foundation of China | 21778051 | Changlin Tian |
| Chinese Academy of Sciences | GJHZ201946 | Changlin Tian |

The funders had no role in study design, data collection and interpretation, or the decision to submit the work for publication.

### Author contributions

Demeng Sun, Data curation, Formal analysis, Funding acquisition, Validation, Methodology, Writing - original draft; Sanling Liu, Siyu Li, Data curation, Validation, Methodology; Mengge Zhang, Fan Yang, Ming Wen, Pan Shi, Data curation, Methodology; Tao Wang, Methodology; Man Pan, Conceptualization, Data curation, Methodology; Shenghai Chang, Resources, Data curation, Methodology; Xing Zhang, Conceptualization, Resources, Data curation; Longhua Zhang, Conceptualization, Data curation, Supervision, Investigation, Writing - review and editing; Changlin Tian, Supervision, Funding acquisition, Investigation, Project administration, Writing - review and editing; Lei Liu, Conceptualization, Supervision, Funding acquisition, Project administration, Writing - review and editing

### Author ORCIDs

Changlin Tian (iD) https://orcid.org/0000-0001-9315-900X

### Decision letter and Author response

Decision letter https://doi.org/10.7554/eLife.57096.sa1
Author response https://doi.org/10.7554/eLife.57096.sa2

## Additional files

### Supplementary files

- Transparent reporting form

## Data availability

The EM maps for hASIC1a and hASIC1a-Mamba1 complex have been deposited in EMDB (www.ebi. ac.uk/pdbe/emdb/) with accession codes EMD-30346 and EMD-30347. The atomic coordinates for hASIC1a and hASIC1a-Mamba1 complex have been deposited in the Protein Data Bank (www.rcsb. org) with accession codes 7CFS and 7CFT respectively.

The following datasets were generated:

| Author(s) | Year | Dataset title | Dataset URL | Database and Identifier |
|---|---|---|---|---|
| Sun DM, Liu SL, Li SY, Yang F, Tian CL | 2020 | Cryo-EM strucutre of human acid-sensing ion channel 1a at pH 8.0 | https://www.rcsb.org/structure/7CFS | RCSB Protein Data Bank, 7CFS |
| Sun DM, Liu SL, Li SY, Yang F, Tian CL | 2020 | Cryo-EM strucutre of human acid-sensing ion channel 1a in complex with snake toxin Mambalgin1 at pH 8.0 | https://www.rcsb.org/structure/7CFT | RCSB Protein Data Bank, 7CFT |

The following previously published datasets were used:

| Author(s) | Year | Dataset title | Dataset URL | Database and Identifier |
|---|---|---|---|---|
| Hoagland EN, Sherwood TW, Lee KG, Walker CJ, Askwith CC | 2010 | Identification of a calcium permeable human acid-sensing ion channel 1 transcript variant | https://www.ncbi.nlm.nih.gov/nuccore/HM991481 | NCBI GenBank, HM991481.1 |
| Yoder N, Yoshioka C, Gouaux E | 2018 | Gating mechanisms of acid-sensing ion channels | https://pubmed.ncbi.nlm.nih.gov/29513651/ | PDB, 6AVE |
| Baconguis I, Bohlen CJ, Goehring A, Julius D, Gouaux E | 2014 | X-ray structure of acid-sensing ion channel 1-snake toxin complex reveals open state of a Na(+)-selective channel | https://pubmed.ncbi.nlm.nih.gov/24507937/ | PDB, 4NTW |

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
