## [Decision Letter]

**Acceptance summary:**

ASIC channels are important proteins gated as a result of extracellular increases in proton concentration. These channels are involved in pain pathways and are considered important pharmacological targets. A toxin derived from the African mamba snake targets ASICs and are useful as tools for structure-function studies.

In this manuscript, Sun et al., present the first structure of the human ASIC1a channel in complex with the toxin Mambalgin 1 (Mamba1) as well as an apo structure. The data show that inhibition of Mamba1 on hASIC is likely a negative allosteric effect on pH-induced conformational changes that normally lead to channel opening. Interestingly, the several residues have been identified that may be responsible for differences in the mode of action of Mamba1 on the human vs. the chicken channel orthologs.

The data is of high quality and after revision, the authors have responded to reviewer's comments and produced a clearly improved manuscript. The structures and accompanying data represent an important contribution to the field, given that these are the first structures of a human ASIC channel.

**Decision letter after peer review:**

Thank you for submitting your article "Structural Insights into Human Acid-sensing Ion Channel 1a Inhibition by Snake Toxin Mambalgin1" for consideration by *eLife*. Your article has been reviewed by 3 peer reviewers, and the evaluation has been overseen by a Reviewing Editor and Richard Aldrich as the Senior Editor. The following individuals involved in review of your submission have agreed to reveal their identity: Leon D Islas (Reviewer #1).

The reviewers have discussed the reviews with one another and the Reviewing Editor has drafted this decision to help you prepare a revised submission.

Summary:

ASIC channels are important proteins that sense and gate as a result of extracellular increases in proton concentration. These channels participate in pain pathways and are considered important pharmacological targets. A toxin derived from the African mamba snake targets ASICs and these toxins have been used as tools for structure function studies.

In this manuscript, Sun et al. present the structure of the human ASIC1a channel in complex with the toxin Mambalgin 1 (Mamba1) as well as an apo structure. The authors show that the mechanism of inhibition of Mamba1 on hASIC is likely a negative allosteric effect on pH-induced conformational changes that normally lead to channel opening. Interestingly, the several residues have been identified that may be responsible for differences in the mode of action of Mamba1 on the human vs. the chicken channel orthologs.

The data is of high quality and the authors have produced a clear manuscript. The structures and accompanying data represent a nice contribution to the field since these are the first structures of a human ASIC channel. The main new finding is the reporting on the first human ASIC1 structure and the authors should emphasize this, especially in the Discussion section.

Essential revisions:

1) Figure 3E and F shows summary data from current inhibition experiments in several ASIC1a-deltaC mutants. It is important to show that the resulting mutations do not alter too much the function of the channel. The inhibition observed could be caused not only by the proposed reduced interaction of Mamba1 with the channel. At least show as complimentary figure a comparison of the mutant's expression level with WT and if available, the extent of activation induced by pH changes.

2) Figure 4E shows a small but significant change in the pore diameters at about a distance of 90, the pore being wider in the presence of Mamba1 toxin. Are there differences in the single channel conductance of the currents activated in the absence and presence of these toxins? Also, please define the reference for the distance that is defined in Figure 4E. Is it distance from the inner leaflet of the membrane?

3) The paper presents the first structure of human ASIC, bound to mambalgin1. As already shown in earlier work from the same group(s), the study reiterates that the thumb domain is the interaction site for mambalgin1 and proposes a possible mechanism of action for the toxin. The manuscript contains insightful electrophysiology data, but novelty and impact are low because the authors (and/or others) have shown similar experiments in the past. Also, the structural basis used to propose the mechanism is questionable, as the local resolution at the mambalgin1 binding site and in the TMs is not sufficient for detailed conclusions.

4) Both key aspects of functional and structural data have been shown previously in Sun et al., 2018 and Diochet et al., 2012. The only key advance of this work is the presentation of a structure of a truncated human ASIC1a construct, which looks extremely similar to the chicken ASIC1 structure. The authors should give special weigh to the fact that this is the first structure of a human ASIC1 channel.

5) The overall resolution of the structures shown here are low, especially in regions critical for mambalgin1 binding and in the lower TM2, where the authors claim to see the largest differences between apo and toxin-bound structures. The authors should make clear what are the limitations on the interpretation of their data imposed by this low resolution.

6) The Discussion section is much more of a summary than a discussion. The authors miss the opportunity to discuss the possible mechanism in the broader context of ion channel modulation, and that of ASICs in particular. This should be improved.

7) The presentation of structural data in almost all figures needs substantial revision to make it more accessible and improve readability.

8) Subsection “Mamba1 Reduces the Proton Sensitivity of hASIC1a”. In the experiment illustrated in Figure 5A, at the end of the co-application, the current is visibly reduced to ~40% of the initial control value. After that, there is additional and longer (!) application of the toxin that leads to additional current reduction. This experiment does not allow comparison of the rates of toxin binding in the presence and absence of activating protons. It does show an additive effect of toxin applications (the rate of current recovery and correspondingly dissociation of the toxin is extremely slow according to the same experiment) but does not support author's conclusion about toxin favoring the resting state. This experiment should be redone in the form where the same duration toxin applications will be administered either in the presence (one experiment) or absence (second experiment) of the stimulating pH 6.0 application. The authors should then measure the current amplitude at the end of such applications and statistically compare them. Based on such comparison, the authors may then make conclusions whether the toxin favors a certain state or not. As of now, the data presented in Figure 5A rather argue that toxin binds hASIC1a indiscriminately of the activation state.

---

## [Author Response]

Summary:ASIC channels are important proteins that sense and gate as a result of extracellular increases in proton concentration. These channels participate in pain pathways and are considered important pharmacological targets. A toxin derived from the African mamba snake targets ASICs and these toxins have been used as tools for structure function studies.In this manuscript, Sun et al. present the structure of the human ASIC1a channel in complex with the toxin Mambalgin 1 (Mamba1) as well as an apo structure. The authors show that the mechanism of inhibition of Mamba1 on hASIC is likely a negative allosteric effect on pH-induced conformational changes that normally lead to channel opening. Interestingly, the several residues have been identified that may be responsible for differences in the mode of action of Mamba1 on the human vs. the chicken channel orthologs.The data is of high quality and the authors have produced a clear manuscript. The structures and accompanying data represent a nice contribution to the field since these are the first structures of a human ASIC channel. The main new finding is the reporting on the first human ASIC1 structure and the authors should emphasize this, especially in the Discussion section.Essential revisions:1) Figure 3E and F shows summary data from current inhibition experiments in several ASIC1a-deltaC mutants. It is important to show that the resulting mutations do not alter too much the function of the channel. The inhibition observed could be caused not only by the proposed reduced interaction of Mamba1 with the channel. At least show as complimentary figure a comparison of the mutant's expression level with WT and if available, the extent of activation induced by pH changes.

We thank the reviewer for pointing this out.

Following the reviewer’s suggestion, we have conducted surface expression level analysis of hASIC1a and hASIC1a mutants in CHO cells, using the flow cytometry method.

As the data shown in Author response image 1, the expression levels of wild-type hASIC1a and hASIC1a mutants are comparable, indicating that the altered inhibitions of hASIC1a mutants by Mamba1 are not caused by the expression levels of the mutants. Meanwhile, we would like to point out that, the expression level of hASIC1a or hASIC1a mutants could not affect the inhibition of the channel by Mamba1 toxin represented in this article. As we have stated in the manuscript, the channel currents were recorded using the single cell patch clamp method. The inhibition effect of Mamba1 on the channel was represented using the ratio of current amplitudes in the presence and absence of the toxin recorded from the same cell membrane fraction. Therefore, the difference in channel expression level between single cells does not affect the measured inhibitory effect of the toxin on the channel.

**Author response image 1. sa2fig1:** Cell surface expression levels of wild-type hASIC1a and hASIC1a mutants. Cell surface expression level of the protein were determined by flow cytometry. The data represent mean ± S.D. from 3 independent experiments. Data were analyzed using one-way analysis of variance and t-test in which each mutant was compared to the wild-type channel (WT). The P values are among 0.8-0.9, indicating that the cell surface expression of each hASIC1a mutant is not significantly different from the wild-type hASIC1a.

We have also analyzed the pH dose-response for the activation of wild type hASIC1a and hASIC1a mutants. The pH-dependence curves of mutants E364A, Q358A and Y360A do not shift compared to the wild-type hASIC1a (Figure 3—figure supplement 4), indicating that the mutations do not alter too much the extent of activation induced by pH changes. While the mutants D347A, D351G and F352L have their pH-dependence curves shift toward more acidic pH (Figure 3—figure supplement 4). Moreover, we found that the presence of Mamba1 toxin do not alter the pH-dependence curves of these mutants, but can shift the curve of wild-type hASIC1a (Figure 5C and 5D). These data could support that the inhibition of hASIC1a mutants by Mamba1 observed are caused by the proposed reduced interaction of Mamba1 with the channel, rather than the alteration of the channel function.

2) Figure 4E shows a small but significant change in the pore diameters at about a distance of 90, the pore being wider in the presence of Mamba1 toxin. Are there differences in the single channel conductance of the currents activated in the absence and presence of these toxins? Also, please define the reference for the distance that is defined in Figure 4E. Is it distance from the inner leaflet of the membrane?

Following the reviewer’s suggestion, especially to verify the pore diameter changes of the channel, we have recorded the single channel currents of hASIC1a activated in the absence and presence of Mamba1 toxin. The data shows that the conductance of the current in the absence of Mamba1 (0.405 pS; Author response image 2) is much larger than that in the presence of Mamba1 (0.029 pS; Author response image 2). This observation is in line with our expectations.

**Author response image 2. sa2fig2:** Single-channel currents of hASIC1a. hASIC1a channel was reconstituted in POPC/POPG (3:1) lipid vesicles. The single channel currents of hASIC1a were recorded at pH 6.0 in the absence (A) and presence (B) of Mamba1 toxin respectively. The membrane potentials were set at 60 mV and 100 mV.

Moreover, to investigate the essence of the observed significant changes in pore diameter of apo-hASIC1a^∆C^ versus hASIC1a^∆C^-Mamba1 complex, we have carefully analyzed the maps and structures of apo hASIC1a^∆C^ and hASIC1a^∆C^-Mamba1 complex.

The map of apo hASIC1a showed an extra electron density located above the base of the extracellular vestibule (Author response image 3). Recalling that 2 mM Ca^2+^ was presence in the buffer during purification of hASIC1a^∆C^, a Ca^2+^ ion was manually fit to this density, which was coordinated by six carboxyl oxygen atoms from symmetry-related Asp434 residues (Author response image 3). Nevertheless, we do not have Ca^2+^ present in the buffer during purification hASIC1a^∆C^-Mamba1 complex. Accordingly, no extra density was observed in the extracellular vestibule of the pore in the map of hASIC1a^∆C^-Mamba1 complex. Therefore, we hypothesized that Ca^2+^ could bring close contacts of the three TM2 helices, resulting in a decreased pore diameter.

**Author response image 3. sa2fig3:** The putative Ca^2+^ located in the vestibule of the pore of hASIC1a^∆C^.

About definition of the reference for the distance in Figure 4E, the Y-axis represents the longitudinal distance along the pore, from the inner leaflet of the membrane to the top of extracellular side of the channel.

3) The paper presents the first structure of human ASIC, bound to mambalgin1. As already shown in earlier work from the same group(s), the study reiterates that the thumb domain is the interaction site for mambalgin1 and proposes a possible mechanism of action for the toxin. The manuscript contains insightful electrophysiology data, but novelty and impact are low because the authors (and/or others) have shown similar experiments in the past. Also, the structural basis used to propose the mechanism is questionable, as the local resolution at the mambalgin1 binding site and in the TMs is not sufficient for detailed conclusions.

We appreciate the reviewer’s criticizing comments.

In our previous work (Sun et al., (2018)), we reported the cryo-EM structure of chicken ASIC1 (cASIC1) in complex with Mamba1, and preformed electrophysiology analysis for cASIC1 mutants and Mamba1 mutants based on the structure. The overall resolution of the human ASIC1a channels reported in this article (3.56 Å for hASIC1a^∆C^, 3.90 Å for hASIC1a^∆C^-Mamba1 complex) are higher than that of cASIC1 (5.7 Å). Especially the transmembrane domains can be clearly observed in hASIC1a structure in this manuscript, while not observed in cryo-EM structure of cASIC1 in our previous work.

Although the electrophysiology data in this manuscript are similar with those in our previous paper, the experimental results in this manuscript are essential for verifying the interaction between Mamba1 toxin and human ASIC1a. The data in this manuscript provide direct evidences supporting the structural mechanism underlying the interaction between human ASIC1a channel and Mamba1 toxin.

We agree that the reported resolution of the hASIC1a cryo-EM structures are not high enough to illustrate more detail interactions between toxin-peptide and hASCI1a. However, we do have tried our best to obtain high resolution structures of hASIC1a and hASIC1a-Mamba1 complex through optimizing the sample conditions (detergents, lipodisc, …) and grid lyophilization parameters.

In the process of cryo-EM structure determination, we noticed that the relatively small transmembrane domain (containing only 6 transmembrane helices), and the non-continuous TM2 (bent at GAS motif) are the major obstacles for resolution improvements of the transmembrane domain. Moreover, the thumb domain of hASIC1a in resting or inhibition state was observed to shift away from the core of the trimeric channel, which lead to the unstable conformation of thumb domain and consequent low local resolution at the toxin-channel binding site. Combinations of these limitations might be the major obstacles preventing us to structures reach high resolution of hASIC1a-Mamba1 complex.

Nevertheless, relatively high resolution (reached to 3 Å) was observed at the extracellular core domain of hASIC1a in this manuscript, including the finger, the β-ball and the palm domains. The binding interface between Mamba1 and the thumb domain of hASIC1a, the extended conformation of the acidic pocket of hASIC1a-Mamba1 complex, are supported well by the cryo-EM map. More importantly, the key residues of Mamba1 that mediate toxin-channel interaction could be well assigned. These could provide solid, although not sufficient, evidence for the proposed mechanism of action for the toxin onto the hASIC1a channel. While the detailed contacts between residues and the conformational change in the TMD are waiting for further verification and investigation.

Parallel with the cryo-EM structure determination of hASIC1a and hASIC1a-Mmaba1, we have worked hard to resolve the structures using X-ray crystallography. We do have obtained the crystals of hASIC1a^∆C^ and hASIC1a^∆C^-Mamba1 complex. Unfortunately, the diffraction data of the crystals were quite poor (4.5 Å for the best), preventing us to resolve the structures.

In this manuscript, it is the first time for us to understand the detail inhibition mode of Mamba1 to hASIC1a channel, and this mechanism might be a good reference to decipher interactions of other toxin-peptides to hASIC1a channel. The proposed channel inhibition mechanism of Mamba1 onto hASIC1a is based on the structural data. To make the Discussion section and conclusion more rigorous, we have revised the statements about the proposed mechanism for the toxin inhibiting hASIC1a channel.

4) Both key aspects of functional and structural data have been shown previously in Sun et al., 2018 and Diochet et al., 2012. The only key advance of this work is the presentation of a structure of a truncated human ASIC1a construct, which looks extremely similar to the chicken ASIC1 structure. The authors should give special weigh to the fact that this is the first structure of a human ASIC1 channel.

Thanks for the reviewer’s suggestion.

We agree that the extracellular domain structures of the cASIC1 was reported in our previous manuscript (Sun et al., (2018) ), but the structure of hASIC1a with both of the extracellular and transmembrane domains were only reported in this manuscript.

In a series of high influence structure papers (Jasti et al., 2007; Gonzales et al., 2009; Baconguis and Gouaux, 2012; Baconguis et al., 2014; Yoder et al., 2018), the full-length structure of cASIC1 (including the extracellular and transmembrane domains) were reported in different functional state, but the inhibition mechanism of Mamba1 onto the hASIC1a (rather than cASIC1) was first time reported in this manuscript. Especially, the Mamba1 was considered to have high pharmaceutical potential to relief pains. In Diochet et al., 2012 paper, the interactions between Mamba1 and ASIC channel was well studied using mutations and functional assays, which was complementarily explored in this manuscript through analyzing the cryo-EM structures of hASIC1a and hASIC1a-Mamba complex.

Some special weighs were emphasized in the Discussion section:

Introduction, subsection “The Intradomain Interaction Affects the Mamba1 Inhibition of ASIC”: We emphasized that the structures reported in this article representing the first human ASIC1 structure.

Discussion section: we have compared the channel modulation modes by different toxin peptides, e.g. PcTx1, MitTx or Mamba1.

Discussion section: we have stated a new inhibition hypothesis of “closed-state trapping mechanism” for the mambalgin-1 onto hASIC1a.

Discussion section: we have analyzed and stated that “Mamba1 showed stronger efficiency on hASIC1a than on cASIC1.”. This clearly demonstrated the significance of the structure of hASIC1a and hASIC1a-Mamba1, simply because the pharmaceutical potential on human ASIC1a channel (a drug to human being disease), not the chicken channel. Especially, this is the first report of the cryo-EM structure of human ASIC1a channel, not chicken ASIC1 channel.

5) The overall resolution of the structures shown here are low, especially in regions critical for mambalgin1 binding and in the lower TM2, where the authors claim to see the largest differences between apo and toxin-bound structures. The authors should make clear what are the limitations on the interpretation of their data imposed by this low resolution.

We fully understand the reviewer’s concern on the relative low resolution of the hASIC1a and hASIC1a-Mamba1 structures. We agree that the reported resolution of the hASIC1a cryo-EM structures are not high enough (3.56 Å for hASIC1a^∆C^, 3.90 Å for hASIC1a^∆C^-Mamba1 complex) to illustrate more detail interactions between toxin-peptide and hASCI1a. However, we do have tried our best to obtain high resolution structures of hASIC1a and hASIC1a-Mamba1 complex through optimizing the sample conditions (detergents, lipodisc, …) and grid lyophilization parameters.

In the process of cryo-EM structure determination, we noticed that the relatively small transmembrane domain (containing only 6 transmembrane helices), and the non-continuous TM2 (bent at GAS belt) are the major obstacles for resolution improvements of the transmembrane domain. Moreover, the thumb domain of hASIC1a in resting or inhibition state was observed to shift away from the core of the trimeric channel, which lead to the unstable conformation of thumb domain and consequent low local resolution at the toxin-channel binding site. Combinations of these limitations might be the major obstacles preventing us to structures reach high resolution of hASIC1a-Mamba1 complex.

Nevertheless, relatively high resolution (reached to 3 Å) was observed at the extracellular core domain of hASIC1a in this manuscript, including the finger, the β-ball and the palm domains. The binding interface between Mamba1 and the thumb domain of hASIC1a, the extended conformation of the acidic pocket of hASIC1a-Mamba1 complex, are supported well by the cryo-EM map. More importantly, the key residues of Mamba1 that mediate toxin-channel interaction could be well assigned. These could provide solid, although not sufficient, evidence for the proposed mechanism of action for the toxin onto the hASIC1a channel. While the detailed contacts between residues and the conformational change in the TMD are waiting for further verification and investigation.

To make the Discussion section and conclusion more rigorous, we have made a statement about the limitations on the interpretation of our data imposed by this low-resolution structure in the revised manuscript.

6) The Discussion section is much more of a summary than a discussion. The authors miss the opportunity to discuss the possible mechanism in the broader context of ion channel modulation, and that of ASICs in particular. This should be improved.

We thank the reviewer for pointing out this and the suggestion. We have now discussed this extensively in the revised manuscript.

Discussion section: we have compared the channel modulation modes by different toxin peptides, e.g. PcTx1, MitTx or Mamba1.

Discussion section: we have stated a new inhibition hypothesis of “closed-state trapping mechanism” for the mambalgin-1 onto hASIC1a.

Discussion section: we have clearly analyzed and stated that “Mamba1 showed stronger efficiency on hASIC1a than on cASIC1.”. This clearly demonstrated the significance of the structure of hASIC1a and hASIC1a-Mamba1, simply because the pharmaceutical potential on human ASIC1a channel (a drug to human being disease), not the chicken channel. Especially, this is the first report of the cryoEM structure of human ASIC1a channel, not chicken ASIC1 channel.

7) The presentation of structural data in almost all figures needs substantial revision to make it more accessible and improve readability.

We have redone the structure presentations in all the main figures and supplementary figures in the manuscript revision. Hopefully, these figures improvements could make these figures more accessible and readable.

8) Subsection “Mamba1 Reduces the Proton Sensitivity of hASIC1a”. In the experiment illustrated in Figure 5A, at the end of the co-application, the current is visibly reduced to ~40% of the initial control value. After that, there is additional and longer (!) application of the toxin that leads to additional current reduction. This experiment does not allow comparison of the rates of toxin binding in the presence and absence of activating protons. It does show an additive effect of toxin applications (the rate of current recovery and correspondingly dissociation of the toxin is extremely slow according to the same experiment) but does not support author's conclusion about toxin favoring the resting state. This experiment should be redone in the form where the same duration toxin applications will be administered either in the presence (one experiment) or absence (second experiment) of the stimulating pH 6.0 application. The authors should then measure the current amplitude at the end of such applications and statistically compare them. Based on such comparison, the authors may then make conclusions whether the toxin favors a certain state or not. As of now, the data presented in Figure 5A rather argue that toxin binds hASIC1a indiscriminately of the activation state.

We thank the reviewer for the insightful comments and the kind suggestions.

Following reviewer’s comments, we have redone the experiments, as illustrated in Figure 5.

As shown in the revised Figure 5A, the durations (500 nM Mamba1) of toxin application were administered either in the absence (upper panel) or presence (middle and lower panels) of the stimulating pH 6.0 application. The data illustrate that application of Mamba1 at pH 8.0 (hASIC1a channel in a resting state) could decrease the channel current simulated by pH 6.0 application. Whereas co-application of 500 nM Mamba1 in the presence of the pH 6.0 stimulation could not decrease the channel current, even though prolong the co-application of the toxin and pH 6.0 stimulation.

The current amplitude at the end of such applications were measured and compared statistically, as shown in Figure 5B.

We believe these data could provide more evidences to support our hypothesis that “Mamba1 toxin favoring the resting state of hASIC1a channel”.